# Covariance-Aware Transformers for Quadratic Programming and Decision Making

**Kutay Tire** [* 1] **Yufan Zhang** [* 2] **Ege Onur Taga** [2] **Samet Oymak** [2]

## Abstract

We explore the use of transformers for solving quadratic programs and how this capability benefits decision-making problems that involve covariance matrices. We first show that the linear attention mechanism can provably solve unconstrained QPs by tokenizing the matrix variables (e.g. $A$ of the objective $\frac{1}{2}x^\top Ax + b^\top x$) row-by-row and emulating gradient descent iterations. Furthermore, by incorporating MLPs, a transformer block can solve (i) $\ell_1$-penalized QPs by emulating iterative soft-thresholding and (ii) $\ell_1$-constrained QPs when equipped with an additional feedback loop. Our theory motivates us to introduce *Time2Decide*: a generic method that enhances a time-series foundation model (TSFM) by explicitly feeding the covariance matrix between the variates. We empirically find that *Time2Decide* uniformly outperforms the base TSFM model for the classical portfolio optimization problem that admits an $\ell_1$-constrained QP formulation. Remarkably, *Time2Decide* can also outperform the two-step "Predict-then-Optimize (PtO)" procedure where we first forecast the returns and then explicitly solve a QP. Our results reveal that transformers benefit from explicitly admitting second-order statistics and this can enable them to effectively solve two-step decision-making problems in one forward pass.

## 1. Introduction

Quadratic programming (QP) is a central tool in modern optimization, underlying applications in control, machine learning, and finance. Its canonical form is

$$\min_x \ \frac{1}{2}x^\top Ax + b^\top x \quad \text{subject to} \quad Cx \preceq d,$$

where $A, b$ define the objective and $Cx \preceq d$ encodes constraints. Beyond this template, many practically important variants, including $\ell_1$-regularized and $\ell_1$-constrained formulations, remain convex and capture structural requirements such as sparsity, budgets, and transaction-cost constraints.

We study the following concrete question: *To what extent can transformer architectures act as general-purpose QP solvers, and how does this capability translate into improved decision making when second-order statistics (e.g., covariance matrices) matter?* This focus is motivated by two observations: First, there is growing evidence that transformers can emulate iterative algorithms and optimization procedures through their attention-and-residual structure (e.g., theories of in-context learning showing gradient-descent-like behavior). Second, many high-stakes decision problems are naturally expressed as QPs whose inputs include covariances. However, standard transformer pipelines, specifically time-series foundation models, do not explicitly incorporate such second-order information, even when it is essential for the downstream task.

A standard example is portfolio optimization. Classical mean–variance allocation solves a QP whose parameters depend on forecasted returns and an (estimated) covariance matrix. A fundamental and widely used baseline is the *two-stage* (or *predict-then-optimize*) pipeline: first predict the unknown parameters (e.g., returns), then solve the resulting QP with a conventional optimizer (e.g., Markowitz, 1952; Elmachtoub and Grigas, 2022). While well established, this decoupling can be statistically suboptimal: when forecasts are noisy, the plug-in solution can overreact and yield inferior decisions. Indeed, one can formalize an explicit scenario where the predict-then-optimize (PtO) rule is *strictly dominated* by a Bayes-optimal end-to-end policy that shrinks noisy estimates before optimizing (Prop. I.1).

As our central contribution, we develop a comprehensive theory on how and when transformers can solve regularized QPs, as summarized in Table 3. Our construc-

---

[1]Department of ECE, University of Texas at Austin, USA [2]Department of ECE, University of Michigan, USA. Correspondence to: Kutay Tire <kutaytire@utexas.edu>.

*Proceedings of the 43rd International Conference on Machine Learning*, Seoul, South Korea. PMLR 306, 2026. Copyright 2026 by the author(s).

tions are highly compute-efficient and capture the individual contributions of attention mechanism, MLP, and looping. Building on this theory, we introduce *Time2Decide*, a generic method to endow time-series foundation models (TSFMs) with transformer-based optimization that is *covariance-aware*. Concretely, Time2Decide augments the TSFM input with tokens that explicitly represent the covariance structure, enabling the model to learn an end-to-end mapping from historical context to decisions in one forward pass. This directly targets the limitation highlighted above: by training to predict the eventual decision variable, the model can learn the proper amount of attenuation/calibration in the presence of estimation noise, rather than committing to a brittle plug-in rule. Empirically, Time2Decide consistently improves over (i) the base TSFM and (ii) supervised fine-tuning of the TSFM *without* covariance information, and it can even outperform the classical two-stage PtO procedure that forecasts returns and then solves the QP explicitly. In particular, while PtO is highly competitive, Time2Decide surpasses it in regimes where end-to-end calibration matters most, aligning with the strict separation exhibited by Proposition 5.1 and illustrating how explicitly admitting second-order statistics can enable transformers to solve two-step decision problems *within a single learned model*. Our contributions are:

- **QP theory.** We provide explicit constructions showing how transformer components can emulate classical first-order methods for multiple QP classes, including unconstrained, linearly constrained, $\ell_1$-regularized, and $\ell_1$-constrained programs.

- **Experimental validation.** We demonstrate that transformers can learn the mapping from tokenized QP instances to optimal solutions, with linear attention variants offering strong empirical performance.

- **Method: Time2Decide.** We integrate QP-solving capabilities into TSFMs via covariance-aware tokenization, enabling end-to-end forecasting and decision-making. Time2Decide outperforms fine-tuning without covariance tokens and can outperform the two-stage PtO baseline. Proposition 5.1 provides a setting where such end-to-end decision-making can *provably* and *strictly* dominate two-stage optimization.

Together, our results position transformers as both (i) theoretically grounded algorithm emulators for regularized QPs and (ii) practical end-to-end decision makers when second-order/covariance information is essential.

## 2. Neural QP Solver Experiments

Our theoretical findings demonstrate that a transformer architecture has the expressive power to represent the computational steps of various QP-solving algorithms. In this sec-

tion, we move from theoretical possibility to empirical validation. We conduct a series of experiments to determine if a transformer can learn the complex mapping from a QP's parameters to its optimal solution through data-driven training alone, without being explicitly programmed with an iterative algorithm.

### 2.1. Experimental Design

We conduct a controlled function-approximation study to assess whether transformer architectures can generalize across families of linearly-constrained QPs. The task is to learn a mapping from a tokenized QP instance to its optimal solution. Each instance is of the form:

$$\min_{x \in \mathbb{R}^n} \tfrac{1}{2} x^\top A x + b^\top x \quad \text{s.t.} \quad Cx \preceq d,$$

where the dataset generator samples the problem data ($A \succ 0$, $b$, $C$, and $d$) and computes the ground-truth optimal solution $x^\star$ for supervision.

Our default configuration uses $n=5$ variables and $m=3$ constraints with datasets of 50k/10k/10k train/validation/test instances. Problem coefficients are sampled as: (i) $A$ is constructed as $A = GG^\top + 0.1I$ where $G \sim \mathcal{N}(0, I)$ to ensure it is positive definite; (ii) $b \sim \mathcal{N}(0, I)$; (iii) $C \sim \mathcal{N}(0, I)$; and (iv) $d \sim \text{Uniform}(1, 2)$.

To prepare the data for the models, we tokenize each QP instance into a sequence of length $n+m+3$. This sequence consists of one token for each row of $A \in \mathbb{R}^{n \times n}$ and $C \in \mathbb{R}^{m \times n}$ (total $n + m$ tokens), plus three additional tokens for the vector $b$, a zero-padded version of $d$, and a random initializer $x_{\text{init}}$. Each token is a vector in $\mathbb{R}^n$. More details can be found in Appendix E.1.

**Model architectures and metrics.** We compare two encoder-only transformer variants: a *SoftmaxTransformer* and a *LinearTransformer*. The SoftmaxTransformer employs standard multi-head self-attention with softmax normalization, computing attention weights as softmax($QK^T/\sqrt{d_k}$). In contrast, the LinearTransformer uses a more efficient linear attention mechanism that replaces the softmax operation with element-wise products.

To evaluate these architectures, we perform a hyperparameter sweep over model depth (number of layers) and the number of attention heads. Performance is primarily assessed using the coefficient of determination ($R^2$) and Normalized Mean Squared Error (NMSE) on the solution vector, with Mean Squared Error (MSE) as the training loss.

In addition, we report downstream feasibility via constraint violation and optimality via objective sub-optimality in the supplementary materials. We present the results for the $n = 5, m = 3$ case in the main text and defer broader

*Table 1.* $R^2$ on the QP Function-Approximation Task ($n=5$, $m=3$).

| Number of Layers | SoftmaxTransformer | | | | LinearTransformer | | | |
|---|---|---|---|---|---|---|---|---|
| | Number of Heads | | | | Number of Heads | | | |
| | 1 | 2 | 4 | 8 | 1 | 2 | 4 | 8 |
| 1 | 0.275 | 0.334 | 0.468 | 0.635 | 0.331 | 0.401 | 0.561 | 0.632 |
| 2 | 0.578 | 0.658 | 0.711 | 0.779 | 0.693 | 0.817 | 0.865 | 0.881 |
| 4 | 0.863 | 0.894 | 0.898 | 0.917 | 0.934 | 0.958 | 0.962 | 0.965 |
| 8 | 0.923 | 0.934 | 0.934 | 0.934 | 0.972 | **0.974** | 0.970 | 0.971 |
| 16 | 0.929 | **0.935** | 0.926 | 0.931 | 0.960 | 0.964 | 0.969 | 0.967 |

settings to the supplementary materials.

## 2.2. Experimental Results

**Function-approximation accuracy.** Our experimental results demonstrate that transformers are capable of learning to solve QP problems. The best-performing LinearTransformer (8 layers, 2 heads) and SoftmaxTransformer achieved an $R^2$ of **0.974** and **0.935**, respectively. Notably, there is a performance advantage for the LinearTransformer architecture. As summarized in Table 1, the LinearTransformer consistently outperforms the SoftmaxTransformer across the hyperparameter grid. This finding suggests that linear attention mechanisms are particularly well-suited for the numerical reasoning required in QP function approximation. The superior accuracy of the LinearTransformer is further detailed in Figure 2, which shows the tight distribution of its Normalized Mean Squared Error (NMSE) on the test set.

## 3. Decision Making with Covariance-Aware Transformers

### 3.1. Problem Formulation and Data

We consider a portfolio optimization problem with $m$ assets over $T$ time periods. At time $t$, we observe the vector of asset returns $\mathbf{r}_t \in \mathbb{R}^m$ (where $r_{t,i}$ represents the return of asset $i$ at time $t$) and must choose portfolio weights $\mathbf{s}_t \in \mathbb{R}^m$ (where $s_{t,i}$ represents the fraction of wealth allocated to asset $i$ at time $t$) subject to the constraints:

$$\mathbf{s}_t \succeq \mathbf{0} \quad \text{(no short selling for simplicity),}$$
$$\mathbf{1}^\top \mathbf{s}_t = 1 \quad \text{(allocation weights sum to 1),} \quad (1)$$
$$\|\mathbf{s}_t - \mathbf{s}_{t-1}\|_1 \leq \gamma \quad \text{(transaction cost limit).}$$

The rebalancing constraint $\|\mathbf{s}_t - \mathbf{s}_{t-1}\|_1 \leq \gamma$ restricts the total absolute change in portfolio weights, where $\gamma$ represents the maximum allowed rebalancing budget. This constraint proxies transaction costs and prevents excessive portfolio turnover.

To establish a target for supervised training, we compute an *oracle allocation*, $\mathbf{s}_t^\star$. This oracle represents the ideal decision at each step, calculated by solving a mean-variance optimization problem assuming perfect foresight of the next period's returns, $\mathbf{r}_t$. The objective is to maximize risk-adjusted returns:

$$\mathbf{s}_t^\star = \arg\max_{\mathbf{s}_t} \ \mathbf{s}_t^\top \mathbf{r}_t - \lambda\, \mathbf{s}_t^\top \hat{\boldsymbol{\Sigma}}_t \mathbf{s}_t \quad \text{s.t. (1) holds.}$$

Here, $\hat{\boldsymbol{\Sigma}}_t$ is the empirical covariance matrix estimated from historical returns, and $\lambda = 0.1$ is the risk aversion parameter that balances expected return against portfolio variance. The oracle solution $\mathbf{s}_t^\star$ represents the optimal allocation given perfect knowledge of future returns $\mathbf{r}_t$.

**Data generation.** We synthesize multivariate time series using the Linear Model of Coregionalization (LMC) as described in (Journel and Huijbregts, 2003), $\mathbf{X}_i(t) = \sum_{j=1}^{r} w_{ij} f_j(t)$, where $f_j(t)$ are latent Gaussian processes with varied kernels, $w_{ij}$ are mixing weights, and $r$ sets covariance rank. We then process these time series by scaling with $\epsilon = 0.01$ and clipping to the range $[-0.3, 0.8]$ to simulate asset returns. We generate separate training and evaluation datasets: 300 training series and 30 evaluation series, each with $m = 16$ assets and $T = 1024$ time steps. We use sequence length $L = 96$, forecast horizon $H = 96$, and a stride of 1 for training data augmentation.

### 3.2. Experimental Setup

**Proposed architecture.** There exist many transformer-based multivariate time-series models. Among them, *TimePFN*—adapted from one of the strongest baselines, PatchTST ((Nie et al., 2023))—serves as the backbone of our approach. *Time2Decide* augments TimePFN with *Covariance-Augmented Tokenization* (CAT): we compute the empirical covariance matrix over the normalized input window, map each covariance row to a token via a multi-layer nonlinear projection, prepend the resulting $m$ covariance tokens to the standard patch embeddings, apply a unified positional encoding to the entire sequence, and process them jointly through self-attention. The input sequence concatenates historical returns and allocations (a total of $2m$ channels), enabling the model to reason jointly over

*Table 2.* MSE Performance for Each Allocation Strategy Across Different Rebalancing Budgets

| Strategy | $\gamma = 0.5$ | $\gamma = 0.75$ | $\gamma = 1.0$ | $\gamma = 1.25$ | $\gamma = 1.5$ | $\gamma = 1.75$ | $\gamma = 2.0$ |
|---|---|---|---|---|---|---|---|
| Uniform | 0.0425 | 0.0444 | 0.0479 | 0.0485 | 0.0505 | 0.0539 | 0.0586 |
| Pretrained | 0.0447 | 0.0458 | 0.0500 | 0.0514 | 0.0523 | 0.0567 | 0.0604 |
| Predict-then-Optimize | **0.0245** | **0.0294** | **0.0345** | **0.0376** | 0.0413 | 0.0472 | 0.0553 |
| SFT | 0.0408 | 0.0411 | 0.0413 | 0.0446 | 0.0445 | 0.0515 | 0.0565 |
| Time2Decide | 0.0323 | 0.0346 | 0.0386 | 0.0390 | **0.0410** | **0.0442** | **0.0488** |

past market dynamics and portfolio positions. After encoding, we remove the covariance tokens and pass only the temporal tokens to the output projectors to generate predictions for both returns and allocations.

**Baselines.** We evaluate our proposed model, *Time2Decide*, against a suite of baselines: (i) Oracle (perfect-foresight allocation, the upper bound in performance for any strategy); (ii) Predict-then-Optimize (forecast-then-optimize baseline where TimePFN-predicted returns are fed into a conventional mathematical QP solver to compute allocations under the constraints); (iii) Pretrained (TimePFN used purely as a neural forecaster without covariance tokens); (iv) SFT (supervised fine-tuned TimePFN variant); (v) Uniform (equal-weight baseline).

**Training and Inference.** SFT-TimePFN and Time2Decide are trained with a combined loss that balances return and allocation prediction:

$$\mathcal{L} = \lambda_r \|\hat{\mathbf{r}} - \mathbf{r}\|_2^2 + \|\hat{\mathbf{s}} - \mathbf{s}^\star\|_2^2,$$

where $\lambda_r$ is a weighting hyperparameter. We find the optimal $\lambda_r$ for each model via a hyperparameter sweep.

At inference, to ensure all portfolio constraints are satisfied, the model's raw output $\hat{\mathbf{s}}$ is projected into the feasible set by solving the following QP:

$$\mathbf{s}_t = \arg \min_{\mathbf{x}} \|\mathbf{x} - \hat{\mathbf{s}}\|_2^2$$
$$\text{s.t. } \mathbf{x} \succeq 0, \ \mathbf{1}^\top \mathbf{x} = 1, \ \|\mathbf{x} - \mathbf{s}_{t-1}\|_1 \leq \gamma \quad (2)$$

**Evaluation.** We use a training set of 300 multivariate return series and an evaluation set of 30 series. All experiments use a risk coefficient of $\lambda = 0.1$. We evaluate each strategy across a range of rebalancing budgets $\gamma \in \{0.5, 0.75, 1.0, 1.25, 1.5, 1.75, 2.0\}$. Performance is measured by the Mean Squared Error (MSE) between the model's predicted allocations and the oracle's allocations. Crucially, the MSE is computed on the raw model predictions *before* the feasibility projection, providing a stringent standard for evaluation.

More experimental details can be found in Appendix E.2.

### 3.3. Experimental Results

Table 2 summarizes, for each strategy, the best MSE obtained after selecting the optimal combined-loss weights $\lambda_r$ from our sweep across different rebalancing budgets. The detailed results are provided in the supplementary materials. The results show that Time2Decide outperforms SFT-TimePFN and Pretrained-TimePFN, which do not incorporate covariance information. The strong competitor, Predict-then-Optimize, which relies on exact conventional mathematical solvers, achieves the lowest errors at small $\gamma$ values (e.g., 0.0245 at $\gamma = 0.5$ and 0.0294 at $\gamma = 0.75$). As the rebalancing budget increases, Time2Decide shows superior performance, surpassing Predict-then-Optimize and achieving the best results at higher $\gamma$ values (e.g., 0.0442 at $\gamma = 1.75$, and 0.0488 at $\gamma = 2.0$).

## 4. Conclusion

In this paper, we presented a theoretical and empirical study of how transformer architectures can be explicitly constructed to solve classes of quadratic programs. By carefully designing attention heads and feed-forward modules, we showed that single transformer blocks can emulate gradient descent, primal–dual iterations, and proximal updates for unconstrained, linearly constrained, and sparse QPs. These constructions demonstrate that transformers can effectively replicate the iterative structures of classical optimization algorithms.

On the experimental side, we verified that transformers can learn to solve QPs when trained on tokenized problem instances, with linear attention variants consistently achieving higher accuracy. Extending beyond the original task, we showed that incorporating covariance information as tokens enables end-to-end decision making under structural constraints. Our results highlight that embedding second-order problem data into the transformer input leads to meaningful improvements over baselines.

Overall, our results connect classical optimization methods with transformers, showing that transformers can efficiently emulate iterative solvers. Future directions include extending to broader optimization tasks, integrating with other foundation models or reinforcement learning for decision making, and studying the theoretical boundaries of such algorithmic emulation.

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

*Table 3.* Summary of Transformer-Based Optimization Algorithms

| Optimization Program | Architecture | Emulated Algorithm |
|---|---|---|
| Unconstrained QP | 1 Linear Attention Block | Gradient Descent |
| Linearly Constrained QP | 2 Sequential Attention Blocks | Arrow-Hurwicz Primal-Dual Method |
| $\ell_1$-Regularized QP | 1 Linear Attention Block + FFN | ISTA (Proximal Gradient Descent) |
| $\ell_1$-Constrained QP | 1 Linear Attention Block + FFN with Threshold Loop | Projected Gradient Descent |

## A. Theoretical Results

### A.1. Problem Setup

**Notation.** Let $[p] = \{1, \ldots, p\}$ for an integer $p \geq 1$. Lowercase letters (e.g., $x, y$) denote vectors; uppercase letters (e.g., $A, C$) denote matrices. For $A \in \mathbb{R}^{n \times n}$, let $\lambda_{\max}(A)$ be its largest eigenvalue and $\|A\|_2$ its operator norm. We write $\langle u, v \rangle = u^\top v$, use $\|\cdot\|_2$ for the Euclidean norm, and $\|\cdot\|_1$ for the $\ell_1$ norm. For $u \in \mathbb{R}$, set $[u]_+ = \max\{u, 0\}$ (applied elementwise to vectors). The soft-thresholding map is $\mathcal{S}_\theta(y) = \text{sign}(y) \odot (|y| - \theta)_+$, and $\text{Proj}_{\{\|\cdot\|_1 \leq B\}}$ denotes Euclidean projection onto the $\ell_1$-ball. We use $\text{prox}_\psi(y) = \arg\min_x\{\frac{1}{2}\|x - y\|_2^2 + \psi(x)\}$ and write $L = \lambda_{\max}(A)$.

**Quadratic programs.** We study four convex QPs, each specified by problem data and a stepsize pair $(\gamma, \eta)$:

$$\text{(U)} \quad \min_{x \in \mathbb{R}^n} \tfrac{1}{2} x^\top A x + b^\top x, \tag{3}$$

$$\text{(LC)} \quad \min_{x \in \mathbb{R}^n} \tfrac{1}{2} x^\top A x + b^\top x \ \ \text{s.t.} \ \ Cx \preceq d, \tag{4}$$

$$\text{(R)} \quad \min_{x \in \mathbb{R}^n} \tfrac{1}{2} x^\top A x + b^\top x + \lambda\|x\|_1, \qquad \lambda > 0, \tag{5}$$

$$\text{(C)} \quad \min_{x \in \mathbb{R}^n} \tfrac{1}{2} x^\top A x + b^\top x \ \ \text{s.t.} \ \ \|x\|_1 \leq B, \quad B > 0. \tag{6}$$

Throughout, we assume $A \succ 0$; for (LC) we also assume the feasible set $\{x : Cx \preceq d\}$ is nonempty.

**Algorithms to be emulated.** Let $x_k$ (and, when present, $\lambda_k \succeq 0$) denote the current iterate(s), and write $L = \lambda_{\max}(A)$. We have the following algorithms:

**GD (U)** Unconstrained Gradient Descent:

$$x_{k+1} = x_k - \gamma(Ax_k + b), \qquad 0 < \gamma < \frac{2}{L}. \tag{7}$$

**Arrow–Hurwicz (LC)** For problems with linear constraints:

$$\begin{aligned} x_{k+1} &= x_k - \gamma(Ax_k + b + C^\top \lambda_k), & 0 < \gamma &< \frac{2}{L}, \\ \lambda_{k+1} &= \left[\lambda_k + \eta(Cx_{k+1} - d)\right]_+, & \eta\gamma\|C\|_2^2 &< 1. \end{aligned} \tag{8}$$

**ISTA (R)** Iterative Shrinkage-Thresholding Algorithm for regularization:

$$\begin{aligned} y_k &= x_k - \gamma(Ax_k + b), & 0 < \gamma &\leq \frac{1}{L}. \\ x_{k+1} &= \mathcal{S}_{\gamma\lambda}(y_k) = \text{prox}_{\gamma\lambda\|\cdot\|_1}(y_k), \end{aligned} \tag{9}$$

**PGD with $\ell_1$-projection (C)** Projected Gradient Descent onto the $\ell_1$-ball:

$$\begin{aligned} y_k &= x_k - \gamma(Ax_k + b), & 0 < \gamma &\leq \frac{1}{L}. \\ x_{k+1} &= \text{Proj}_{\{\|x\|_1 \leq B\}}(y_k), \end{aligned} \tag{10}$$

We present explicit weight constructions showing how small transformer fragments implement one iteration of classical first-order methods for QPs. Tokens encode the current iterate $x_k$, the multiplier $\lambda_k$ when needed, and rows of problem data (e.g., $A$, $C$). A residual on the $x$-row propagates the iterate, and repeating these one-step maps yields the standard convergence behavior under the usual step size conditions of the target algorithms. The summary of the required architectures for each algorithm is presented in Table 3.

### A.2. Unconstrained QP via Single-Block Linear Attention

We show that a single linear-attention head with a residual connection simulates one step of gradient descent for the unconstrained QP defined in Section A.1.

**Proposition A.1.** *A single linear-attention head that attends from the $x$-row to the $A$-rows, followed by a residual on the $x$-row, realizes the update $x_{k+1} = x_k - \gamma(Ax_k + b)$. Hence, for any $0 < \gamma < 2/L$, the iterates converge linearly to $x^\star = -A^{-1}b$.*

*Construction.* We store tokens for the rows of $A$, the vector $b$, and the current iterate $x_k$, and we design fixed query, key, and value maps $W_Q, W_K, W_V$ (see Appendix C.1). Then the resulting $q, k, v$ yield, when evaluated at the $x$-token is $o = Ax_k + b$. Applying a residual with post-map $-\gamma I$ on the $x$-row yields

$$x_{k+1} = x_k - \gamma\, o = x_k - \gamma(Ax_k + b).$$

Full token layout and selector matrices are given in the Appendix C.1. Furthermore, we empirically validate this construction in Figure 1a and discuss the algorithmic complexity at Appendix D.1. □

### A.3. Linearly Constrained QP via Single Macro-Block Linear Attention

We now turn to the linearly constrained QP, the second formulation presented in Section A.1.

**Proposition A.2.** *A single macro-block consisting of two self-attention blocks and a residual on the $x$-row realizes the sequential update*

$$x_{k+1} = x_k - \gamma\big(Ax_k + b + C^\top \lambda_k\big), \tag{11}$$

$$\lambda_{k+1} = \big[\lambda_k + \eta\,(Cx_{k+1} - d)\big]_+. \tag{12}$$

*Let $L = \lambda_{\max}(A)$ and let $\|\cdot\|_2$ denote the operator norm. If the feasible set $\{x : Cx \preceq d\}$ is nonempty and the step sizes satisfy $0 < \gamma < 2/L$ and $\eta\gamma\|C\|_2^2 < 1$, then the iterates $(x_k, \lambda_k)$ converge to a KKT point.*

*Construction.* We fix $W_Q, W_K, W_V$ so that two heads in the first linear-attention block, evaluated at the $x$- and $\lambda$-tokens, produce

$$o^{Ax+b} = Ax_k + b, \qquad o^{C^\top \lambda} = C^\top \lambda_k,$$

which yields the primal update

$$x_{k+1} = x_k - \gamma\big(o^{Ax+b} + o^{C^\top \lambda}\big) = x_k - \gamma\big(Ax_k + b + C^\top \lambda_k\big).$$

We then run a second attention block with a single head evaluated at $x_{k+1}$, obtaining

$$o^{Cx-d} = Cx_{k+1} - d,$$

and apply a token-wise ReLU for the dual step,

$$\lambda_{k+1} = \big[\lambda_k + \eta\, o^{Cx-d}\big]_+.$$

Full $W_Q, W_K, W_V$ specifications and calculations of the linear attention blocks appear in Appendix C.2, together with the per-iteration cost analysis in Appendix D.2. □

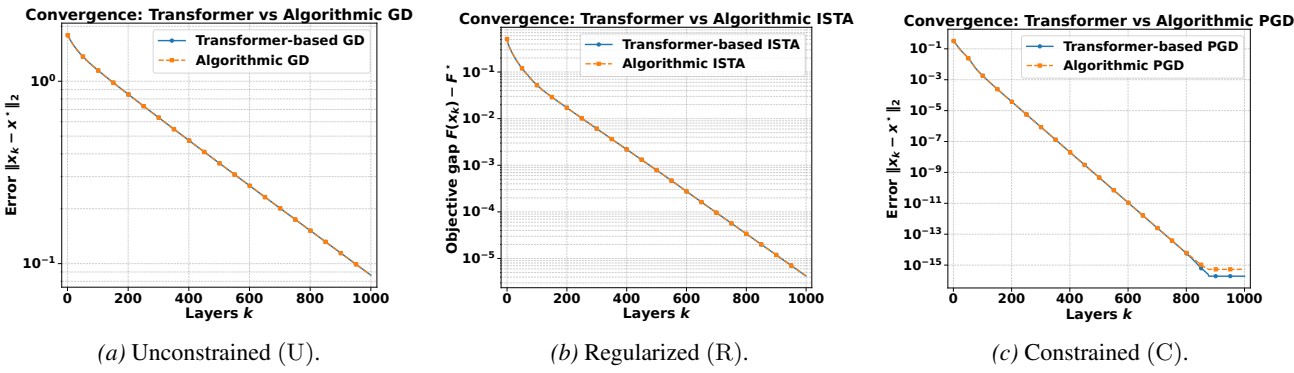

*Figure 1.* Convergence on a semilog-$y$ scale. Each panel compares a standard algorithm (orange) with our transformer-based construction (blue): **(a)** GD for the unconstrained case (U), **(b)** ISTA for the regularized case (R), and **(c)** PGD for the constrained case (C). The curves nearly overlap across layers (depth = iterations), showing that the transformer mirrors the reference methods.

### A.4. Sparse QPs via Single-Block Linear Attention: $\ell_1$-Regularized and $\ell_1$-Constrained

We now introduce sparsity via an $\ell_1$ penalty (R) or an $\ell_1$ budget (C)—corresponding to the last two QP formulations in Section A.1—and extend the (U) block, without changing the attention mechanism, to realize proximal and projected gradient descent algorithms.

**Proposition A.3.** *One (U) gradient step followed by a fixed two-layer ReLU FFN with a scalar threshold $\theta = \gamma\lambda$ yields*

$$x_{k+1} = \mathcal{S}_\theta\big(x_k - \gamma(Ax_k + b)\big) = \mathrm{prox}_{\gamma\lambda\|\cdot\|_1}(y_k).$$

*With $0 < \gamma \leq 1/L$, the objective $F(x) = \frac{1}{2}x^\top Ax + b^\top x + \lambda\|x\|_1$ decreases and $x_k$ converges to a minimizer.*

*Construction.* We compute $y_k = x_k - \gamma(Ax_k + b)$ with the (U) head and implement soft-thresholding with a fixed two-layer width-$2n$ ReLU FFN with weights

$$W_1 = \begin{bmatrix} I_n \\ -I_n \end{bmatrix} \in \mathbb{R}^{2n\times n}, \qquad W_2 = \begin{bmatrix} I_n & -I_n \end{bmatrix} \in \mathbb{R}^{n\times 2n},$$

and bias $-\theta[\mathbf{1}_n; \mathbf{1}_n]$. Then

$$x_{k+1} = W_2\,\mathrm{ReLU}\Big(W_1 y_k - \theta\begin{bmatrix}\mathbf{1}_n\\\mathbf{1}_n\end{bmatrix}\Big)$$
$$= (y_k - \theta\mathbf{1}_n)_+ - (-y_k - \theta\mathbf{1}_n)_+ = \mathcal{S}_\theta(y_k).$$

since $(u - \theta)_+ - (-u - \theta)_+ = \mathrm{sign}(u)\,(|u| - \theta)_+$ coordinatewise. With $\theta = \gamma\lambda$, this equals $\mathrm{prox}_{\gamma\lambda\|\cdot\|_1}(y_k)$, i.e., the ISTA update. Full steps are provided in Appendix C.3. Figure 1b shows that the transformer construction matches ISTA's convergence behavior across depth. □

**Proposition A.4.** *One (U) gradient step followed by the same fixed two-layer ReLU FFN wrapped in a scalar threshold loop performs the exact Euclidean projection onto the $\ell_1$-ball:*

$$x_{k+1} = \mathrm{Proj}_{\{\|x\|_1 \leq B\}}\big(x_k - \gamma(Ax_k + b)\big).$$

*With $0 < \gamma \leq 1/L$, this is projected gradient descent for (C), so $x_k$ converges to an optimal solution.*

*Construction.* From the (U) head we form $y_k = x_k - \gamma(Ax_k + b)$. We then run a scalar threshold loop

$$\theta_{t+1} = \theta_t + \eta\,\big[\|\mathcal{S}_{\theta_t}(y_k)\|_1 - B\big]_+, \qquad \theta_0 = 0, \;\; 0 < \eta \leq \tfrac{1}{n},$$

and set $x_{k+1} = \mathcal{S}_{\theta^\star}(y_k)$ with $\theta^\star = \lim_t \theta_t$. This yields the exact Euclidean projection $x_{k+1} = \mathrm{Proj}_{\{\|x\|_1 \le B\}}(y_k)$. Full calculation is given in Appendix C.4. We empirically verify that the explicit transformer constructions mirror the corresponding first-order iterations across depth. Figure 1 compares GD/ISTA/PGD with their transformer counterparts for (U), (R), and (C), respectively, showing near-overlapping convergence curves. Finally, we provide a corresponding empirical check for the $\ell_1$-constrained case in Figure 1c.

$\square$

## B. Related Work

**Theoretical Analysis of In-Context Learning**   Recent work has developed theoretical frameworks for understanding in-context learning in transformers. Akyürek et al. (2023), Von Oswald et al. (2023) and Dai et al. (2023) demonstrated that transformers emulate gradient descent during ICL. Xie et al. (2022) offered a Bayesian perspective, while Zhang et al. (2023; 2024) showed transformers learn linear models in-context. Ahn et al. (2023) established that they implement preconditioned gradient descent, and Mahankali et al. (2024) proved that one-step gradient descent is optimal for single-layer linear attention. Multiple works (Li et al., 2023; Yang et al., 2024; Li et al., 2024; Bai et al., 2023; Shen et al., 2024) studied the generalization capability of transformers. However, these exclusively focus on fully-supervised settings, leaving a critical gap in understanding how transformers handle partially labeled data—a common real-world scenario. Our work addresses this gap by providing the first theoretical characterization of semi-supervised in-context learning. (Wang et al., 2024) considers a setting where the model observes demonstrations of the form (query, response$_i$, reward$_i$) and aims to correct its response based on the reward sequence. Our work has a different focus as it highlights that the model can correct/impute the missing labels using implicit feedback from labeled demonstrations.

**Neural Networks as Optimizers.**   In recent research, neural networks have been harnessed as optimizers in several directions. (Villarrubia et al., 2018) showed that multilayer perceptrons can approximate non-linear objective functions and transform them into polynomial forms so that classical methods apply when Lagrange multipliers are impractical. Differentiable layers such as (Amos and Kolter, 2017) treat a quadratic program as a network module, derive gradients via implicit differentiation, and use a custom interior-point solver, while (Magoon et al., 2024) decouples gradient computation from the choice of QP solver by exploiting the active set of constraints. In (Nair et al., 2020), neural heuristics improve mixed-integer solvers by learning variable assignments and branching policies. (Chen and Liu, 2022) treats the objective itself as a neural network and optimizes it via backpropagation, handling multi-objective cases with little sensitivity to variable dimension. (Chen et al., 2022) showed that graph neural networks can predict feasibility, boundedness, and approximate solutions for linear programs. Our study differs by feeding quadratic programs directly into transformers and proving that with appropriate tokenization, these sequence models can emulate iterative solvers while handling linear and L1 constraints. Beyond demonstrating feasibility, we provide a systematic analysis of how transformer attention mechanisms capture the structure of quadratic programs, and we show empirically that our model achieves competitive accuracy. In this way, our work establishes transformers themselves as standalone, general-purpose optimizers for constrained quadratic programs.

**Multivariate Time-Series Transformers.**   Early neural approaches to multivariate forecasting relied on encoder–decoder RNNs and temporal convolutional networks; however, their receptive fields or recurrence limited long-range context. Transformer variants now dominate the landscape. Temporal Fusion Transformer (TFT) augments attention with gating and interpretable variable selection (Lim et al., 2021). Informer introduces ProbSparse self-attention for efficient long-sequence inference (Zhou et al., 2021), while Autoformer and FEDformer decompose series into trend and seasonal tokens to improve long-horizon extrapolation (Wu et al., 2021; Zhou et al., 2022). Crossformer (Zhang and Yan, 2023) and Spacetimeformer (Grigsby et al., 2021) rearrange inputs to model cross-dimension dependencies explicitly, and PatchTST (Nie et al., 2023) treats fixed-length segments as tokens to boost locality. Most recently, TimePFN uses permutation-equivariant networks pre-trained on synthetic data to provide strong zero-shot generalization (Taga et al., 2025). These advances demonstrate the transformer's flexibility in processing diverse token structures and incorporating auxiliary information, setting the foundation for integrating optimization-solving capabilities alongside forecasting tasks in multivariate time-series models.

# C. Proofs of Main Results from Section 4

## C.1. Full Proof of the Construction for Unconstrained QP

*Construction.* At iteration $k$, we assemble the token matrix

$$Z_k = \left[\tilde{a}_1^\top; \ldots; \tilde{a}_n^\top; \tilde{b}^\top; \tilde{x}_k^\top\right] \in \mathbb{R}^{(n+2)\times(2n+1)},$$

where

$$\tilde{a}_i^\top = [a_i^\top \ e_i^\top \ 0], \qquad \tilde{b}^\top = [0_n^\top \ b^\top \ 1], \qquad \tilde{x}_k^\top = [x_k^\top \ 0_n^\top \ 1].$$

Here $a_i^\top$ is the $i$-th row of $A$ and $\{e_i\}_{i=1}^n$ are standard basis vectors.

We split each token into *content* vs. *ID* using

$$E = [I_n \ 0_{n\times(n+1)}] \in \mathbb{R}^{n\times(2n+1)}, \qquad \mathbf{R} = [0_{n\times n} \ I_n \ 0_{n\times 1}] \in \mathbb{R}^{n\times(2n+1)},$$

and we expose a trailing constant via $S = [0_{1\times(2n)} \ 1]$.

We use a single linear-attention head with

$$W_Q = [E^\top \ S^\top], \qquad W_K = [E^\top \ S^\top], \qquad W_V = \mathbf{R}^\top.$$

Then the $x$-token query is $q = \tilde{x}_k W_Q = [x_k; 1]$. For $A$-rows, $\tilde{a}_i W_K = [a_i; 0]$ and $\tilde{a}_i W_V = e_i$; for the $b$-token, $\tilde{b} W_K = [0; 1]$ and $\tilde{b} W_V = b$. Evaluating the head at the $x$-token,

$$o = \sum_{i=1}^n \langle[x_k; 1], [a_i; 0]\rangle e_i + \langle[x_k; 1], [0; 1]\rangle b = \sum_{i=1}^n (a_i^\top x_k)\, e_i + b = Ax_k + b.$$

Applying a residual on the $x$-row with post-map $-\gamma I_n$ yields

$$x_{k+1} = x_k - \gamma(Ax_k + b).$$

$\square$

## C.2. Full Proof of the Construction for Linearly Constrained QP

*Construction.* We add a dual token $\lambda_k \in \mathbb{R}_{\geq 0}^m$. As in the unconstrained case, we split tokens into *content* and *ID* parts and embed every token to a common width $p := 2n + m$ (zero-padded), then append a constant slot to reach $p' := p + 1$. Let

$$S = \begin{bmatrix} 0_{1\times n} & 0_{1\times n} & 0_{1\times m} & 1 \end{bmatrix} \in \mathbb{R}^{1\times p'}.$$

Define fixed selectors

$$E_1 = \begin{bmatrix} I_n & 0_{n\times n} & 0_{n\times m} & 0 \end{bmatrix} \in \mathbb{R}^{n\times p'}, \qquad \mathbf{R}_1 = \begin{bmatrix} 0_{n\times n} & I_n & 0_{n\times m} & 0 \end{bmatrix} \in \mathbb{R}^{n\times p'},$$
$$E_2 = \begin{bmatrix} 0_{m\times n} & 0_{m\times n} & I_m & 0 \end{bmatrix} \in \mathbb{R}^{m\times p'}, \quad \mathbf{R}_2 = \begin{bmatrix} 0_{m\times n} & 0_{m\times n} & I_m & 0 \end{bmatrix} \in \mathbb{R}^{m\times p'}.$$

Tokens are

$$\hat{a}_i^\top = \begin{bmatrix} a_i^\top & e_i^{(n)\top} & 0_m^\top & b_i \end{bmatrix}, \quad \hat{c}_i^\top = \begin{bmatrix} c_i^\top & 0_n^\top & e_i^{(m)\top} & -d_i \end{bmatrix},$$
$$\hat{x}_k^\top = \begin{bmatrix} x_k^\top & 0_n^\top & 0_m^\top & 1 \end{bmatrix}, \qquad \hat{\lambda}_k^\top = \begin{bmatrix} 0_n^\top & 0_n^\top & \lambda_k^\top & 1 \end{bmatrix}.$$

We use three fixed (linear) attention heads with $W_Q, W_K, W_V$ chosen as below.

**Head HA+b (evaluated at $x_k$).** Queries/keys/values:

$$q = \hat{x}_k \begin{bmatrix} E_1^\top \\ S^\top \end{bmatrix} = [x_k; 1], \quad k_i = \hat{a}_i \begin{bmatrix} E_1^\top \\ S^\top \end{bmatrix} = [a_i; b_i], \quad v_i = \hat{a}_i \mathbf{R}_1^\top = e_i^{(n)}.$$

Output:

$$o^{Ax+b} = \sum_{i=1}^n \langle[x_k; 1], [a_i; b_i]\rangle\, e_i^{(n)} = Ax_k + b \in \mathbb{R}^n.$$

**Head HC$^\top$ (evaluated at $\lambda_k$).**

$$q = \hat{\lambda}_k E_2^\top = \lambda_k, \quad k_i = \hat{c}_i \, \mathbf{R}_2^\top = e_i^{(m)}, \quad v_i = \hat{c}_i E_1^\top = c_i,$$

so $o^{C^\top \lambda} = \sum_{i=1}^m (\lambda_k)_i c_i = C^\top \lambda_k \in \mathbb{R}^n$.

**Head HC-d (evaluated at $x$).**

$$q = \hat{x} \begin{bmatrix} E_1^\top \\ S^\top \end{bmatrix} = [x; 1], \quad k_i = \hat{c}_i \begin{bmatrix} E_1^\top \\ S^\top \end{bmatrix} = [c_i; -d_i], \quad v_i = \hat{c}_i \, \mathbf{R}_2^\top = e_i^{(m)},$$

so $o^{Cx-d} = \sum_{i=1}^m \langle [x; 1], [c_i; -d_i] \rangle e_i^{(m)} = Cx - d \in \mathbb{R}^m$.

**Two-block macro.** We first compute the two head outputs at the $x$- and $\lambda$-tokens, $o^{Ax+b} = Ax_k + b$ and $o^{C^\top \lambda} = C^\top \lambda_k$, concatenate them, and apply the linear map $W_O^{(1)} = [-\gamma I_n \quad -\gamma I_n]$ with a residual on the $x$-row. This yields

$$x_{k+1} = x_k - \gamma \big( Ax_k + b + C^\top \lambda_k \big).$$

For the second block, using the updated $x_{k+1}$, we evaluate the head $o^{Cx-d} = Cx_{k+1} - d$, scale it with $W_O^{(2)} = \eta I_m$, and perform a token-wise ReLU to enforce non-negativity:

$$\lambda_{k+1} = \big[ \lambda_k + \eta \, (Cx_{k+1} - d) \big]_+.$$

Together, these two blocks implement exactly the Arrow–Hurwicz (projected primal–dual) step of Proposition A.2.

$\square$

### C.3. Full Proof of the Construction for L1-Regularized QP

*Construction.* We compute $y_k = x_k - \gamma(Ax_k + b)$ with the (U) head and implement soft-thresholding with a fixed two-layer width-$2n$ ReLU FFN with weights

$$W_1 = \begin{bmatrix} I_n \\ -I_n \end{bmatrix} \in \mathbb{R}^{2n \times n}, \qquad W_2 = \begin{bmatrix} I_n & -I_n \end{bmatrix} \in \mathbb{R}^{n \times 2n}, \qquad h = W_1 y_k - \theta \begin{bmatrix} \mathbf{1}_n \\ \mathbf{1}_n \end{bmatrix},$$

Let us write $h = \begin{bmatrix} h^+ \\ h^- \end{bmatrix}$ with

$$h^+ = y_k - \theta \mathbf{1}, \qquad h^- = -y_k - \theta \mathbf{1}.$$

After the ReLU we have

$$r = ReLU(h) = \begin{bmatrix} ReLU(h^+) \\ ReLU(h^-) \end{bmatrix} = \begin{bmatrix} (y_k - \theta \mathbf{1}_n)_+ \\ (-y_k - \theta \mathbf{1}_n)_+ \end{bmatrix}.$$

Applying the second linear layer gives

$$x_{k+1} = W_2 r = (y_k - \theta \mathbf{1}_n)_+ - (-y_k - \theta \mathbf{1}_n)_+.$$

Coordinatewise, for each $i \in [n]$,

$$x_{k+1,i} = \max\{y_{k,i} - \theta, 0\} - \max\{-y_{k,i} - \theta, 0\}.$$

A short case analysis shows

$$x_{k+1,i} = \begin{cases} y_{k,i} - \theta, & y_{k,i} > \theta, \\ 0, & |y_{k,i}| \leq \theta, \\ y_{k,i} + \theta, & y_{k,i} < -\theta, \end{cases} \iff x_{k+1,i} = \mathrm{sign}(y_{k,i}) \left( |y_{k,i}| - \theta \right)_+ = \mathcal{S}_\theta(y_{k,i}).$$

Thus, in vector form,

$$x_{k+1} = W_2 \, ReLU\Big( W_1 y_k - \theta \begin{bmatrix} \mathbf{1}_n \\ \mathbf{1}_n \end{bmatrix} \Big) = \mathcal{S}_\theta(y_k) = \mathrm{prox}_{\gamma\lambda\|\cdot\|_1}(y_k),$$

with $\theta = \gamma\lambda$. This realizes the ISTA update we stated. $\square$

## C.4. Full Proof of the Construction for L1-Constrained QP

*Construction.* We reuse the (U) head to compute $y_k = x_k - \gamma(Ax_k + b)$. The fixed two-layer ReLU FFN from Proposition A.3 implements soft-thresholding, so

$$x_t = \mathcal{S}_{\theta_t}(y_k), \qquad s_t = \|x_t\|_1.$$

We drive the scalar threshold toward the $\ell_1$-budget by

$$\theta_{t+1} = \theta_t + \eta\,\mathrm{ReLU}(s_t - B), \qquad \theta_0 = 0,\ \ 0 < \eta \le \frac{1}{n}.$$

Let $s(\theta) = \|\mathcal{S}_\theta(y_k)\|_1 = \sum_{i=1}^n (|y_{k,i}| - \theta)_+$ and $r_t := \mathrm{ReLU}(s_t - B) \ge 0$. On any interval where the active set $A(\theta) = \{i : |y_{k,i}| > \theta\}$ is fixed with size $m \ge 1$,

$$s(\theta) = \sum_{i \in A(\theta)} |y_{k,i}| - m\,\theta, \quad \Rightarrow \quad r_{t+1} = s(\theta_t + \eta r_t) - B \le (1 - m\eta)\,r_t.$$

Thus $0 < \eta \le 1/n$ yields $r_{t+1} \in [0, r_t]$ and $r_t \downarrow 0$; at breakpoints $s$ is nonincreasing, so monotonicity persists. Hence $\theta_t \to \theta^\star \ge 0$ with $s(\theta^\star) = \min\{B, \|y_k\|_1\}$, and we return

$$x_{k+1} = \mathcal{S}_{\theta^\star}(y_k).$$

Finally, $\mathcal{S}_{\theta^\star}(y_k)$ is the unique solution of $\min_x \frac{1}{2}\|x - y_k\|_2^2$ s.t. $\|x\|_1 \le B$ by the KKT conditions $0 \in x - y_k + \theta^\star \partial \|x\|_1$, $\theta^\star \ge 0$, $\theta^\star(\|x\|_1 - B) = 0$. Therefore, the loop computes the exact Euclidean projection onto the $\ell_1$-ball. $\qquad\square$

# D. Computational Cost per Iteration

Our constructions implement *one solver iteration per transformer layer*. Therefore, it is natural to compare the cost of a single transformer step to the corresponding first-order iteration. We organize our analysis by problem class: unconstrained and $\ell_1$-penalized QPs (Subsection D.1), which depend only on the variable dimension $n$, and linearly constrained QPs (Subsection D.2), which depend on both $n$ and the number of constraints $m$. We report the multiplicative overhead:

$$\text{Overhead} = \frac{\text{Transformer}}{\text{Classical}}.$$

### D.1. Per-Iteration Cost for (U), (R), and (C)

These three problem classes—corresponding to Propositions A.1, A.3, and A.4—involve only the quadratic matrix $A \in \mathbb{R}^{n \times n}$ and vectors in $\mathbb{R}^n$, with no explicit linear inequality constraints. The computational cost depends solely on the variable dimension $n$.

**Asymptotic complexity.**    For dense QPs, both classical GD/ISTA and our transformer steps are dominated by the matrix–vector product $Ax$, which costs $O(n^2)$ time per iteration. ISTA additionally applies soft-thresholding at $O(n)$ cost, negligible relative to $n^2$ in the dense regime. For $\ell_1$-constrained PGD, the classical exact projection onto the $\ell_1$-ball runs in $O(n \log n)$ time, whereas our threshold-loop projection costs $O(Tn)$ for $T$ inner updates; the overall step remains dominated by the $O(n^2)$ product.

**Wall-clock time per step.**    Table 4 reports per-step timing for these three problem classes.

**Discussion.**    For Gradient Descent and ISTA, the transformer step achieves near iteration-level parity with overheads of 1.04–1.35× across dimensions. Projected Gradient Descent exhibits larger overhead at small $n$ due to the $O(Tn)$ threshold-loop projection, but this decreases from 3.37× at $n{=}16$ to 1.14× at $n{=}128$ as the $O(n^2)$ matrix–vector product dominates.

*Table 4.* Per-step timing (ms) for unconstrained QP (U), $\ell_1$-regularized QP (R), and $\ell_1$-constrained QP (C). These problems depend only on the variable dimension $n$.

| $n$ | Gradient Descent (U) | | | ISTA (R) | | | Projected GD (C) | | |
|---|---|---|---|---|---|---|---|---|---|
| | Class. | Transf. | Overhead | Class. | Transf. | Overhead | Class. | Transf. | Overhead |
| 16 | 0.040 | 0.050 | 1.25× | 0.052 | 0.070 | 1.35× | 0.108 | 0.364 | 3.37× |
| 32 | 0.286 | 0.310 | 1.08× | 0.305 | 0.341 | 1.12× | 0.416 | 0.696 | 1.67× |
| 64 | 0.327 | 0.341 | 1.04× | 0.358 | 0.389 | 1.09× | 0.700 | 0.970 | 1.39× |
| 128 | 0.595 | 0.642 | 1.08× | 0.658 | 0.719 | 1.09× | 1.452 | 1.653 | 1.14× |

## D.2. Per-Iteration Cost for (LC)

The linearly constrained problem class—corresponding to Proposition A.2—involves explicit linear inequality constraints $Cx \preceq d$ with constraint matrix $C \in \mathbb{R}^{m \times n}$. The Arrow–Hurwicz primal-dual method requires two sequential attention blocks per iteration: one for the primal update and one for the dual update. The computational cost depends on both the variable dimension $n$ and the number of constraints $m$.

**Asymptotic complexity.** Each Arrow–Hurwicz iteration requires forming $Ax$ at $O(n^2)$ cost, plus $Cx$ and $C^\top \lambda$ at $O(mn)$ cost each, giving $O(n^2 + mn)$ per step. The ReLU operation for dual feasibility adds $O(m)$ cost, which is negligible.

**Wall-clock time per step.** Table 5 reports per-step timing for various $(n, m)$ configurations.

*Table 5.* Per-step timing (ms) for linearly constrained QP (LC) using Arrow–Hurwicz iterations. This problem depends on both the variable dimension $n$ and the number of constraints $m$.

| $n$ | $m$ | Classical (ms) | Transformer (ms) | Overhead |
|---|---|---|---|---|
| 16 | 8 | 0.136 | 0.169 | 1.25× |
| 16 | 16 | 0.178 | 0.210 | 1.18× |
| 32 | 8 | 0.543 | 0.580 | 1.07× |
| 32 | 16 | 1.241 | 1.263 | 1.02× |
| 32 | 32 | 1.316 | 1.336 | 1.02× |
| 64 | 16 | 1.339 | 1.388 | 1.04× |
| 64 | 32 | 1.448 | 1.432 | 0.99× |
| 64 | 64 | 1.499 | 1.572 | 1.05× |
| 128 | 32 | 2.282 | 2.712 | 1.19× |
| 128 | 64 | 2.911 | 3.501 | 1.20× |

**Discussion.** The Arrow–Hurwicz transformer construction achieves overheads ranging from $0.99\times$ to $1.25\times$ across all tested configurations. Notably, at $(n, m) = (64, 32)$, the transformer construction is *faster* than the classical implementation, likely due to improved memory access patterns from the unified token representation $Z = [Q; C; c; d; x; \lambda]$. As problem size increases, overheads remain modest (1.19–1.20× at $n{=}128$), demonstrating that the two-block attention structure required for primal-dual updates does not introduce significant computational burden beyond the single-block constructions.

# E. Experimental Details

In this section, we provide comprehensive details regarding the setup of our experiments to ensure reproducibility. All experiments are implemented in PyTorch and executed on NVIDIA L40S GPUs.

## E.1. Neural QP Solver

### E.1.1. QP DATASET GENERATION

We generate synthetic QP instances of the form $\min_x \frac{1}{2} x^\top A x + b^\top x$ s.t. $Cx \preceq d$. The symmetric positive definite matrix $A \in \mathbb{R}^{n \times n}$ is constructed as $A = GG^\top + 0.1I$, where entries of $G \in \mathbb{R}^{n \times n}$, as well as $b \in \mathbb{R}^n$ and constraint matrix $C \in \mathbb{R}^{m \times n}$, are sampled from a standard normal distribution $\mathcal{N}(0, 1)$. The bound vector $d \in \mathbb{R}^m$ is sampled from the uniform distribution $\mathcal{U}[1, 2]$. Ground truth solutions $x^*$ are computed using the OSQP solver accessed via CVXPY. For problem dimension configurations $(n, m) \in \{(5, 3), (7, 3), (7, 6), (10, 3), (10, 6), (10, 9)\}$, we generate a dataset consisting of $50,000$ training samples, $10,000$ validation samples, and $10,000$ test samples. For larger instances with configurations $(n, m) \in \{(15, 10), (20, 8), (20, 16)\}$, we increase the dataset size to $100,000$ training samples, $20,000$ validation samples, and $20,000$ test samples to ensure sufficient coverage of the higher-dimensional space.

### E.1.2. MODEL ARCHITECTURES

We evaluate two architectures: `SoftmaxTransformer`, which uses standard scaled dot-product attention, and `LinearTransformer`, which employs the linear attention mechanism. Both models project the input problem data (tokenized rows of $A, C$ and vectors $b, d$) into a hidden embedding dimension of $d_{\text{model}} = 256$. We conduct a grid search over model depth $L \in \{1, 2, 4, 8, 16, 32\}$ and the number of attention heads $H \in \{1, 2, 4, 8\}$ to analyze the impact of model capacity. A dropout rate of $0.1$ is applied during training to prevent overfitting.

### E.1.3. TRAINING

All models are trained to minimize the Mean Squared Error (MSE) between the predicted $x$ and the ground truth optimal solution $x^*$. We use the AdamW optimizer with an initial learning rate of $1 \times 10^{-4}$ and a weight decay of $0.02$. We utilize a learning rate scheduler (`ReduceLROnPlateau`) that reduces the learning rate by a factor of $0.5$ if the validation loss does not improve for 5 epochs, with a minimum learning rate of $10^{-6}$. The random seed is fixed at $42$ for all experiments to ensure reproducibility. Training proceeds for a maximum of $500$ epochs. To prevent overfitting, we employ early stopping with a patience of 30 epochs, monitoring the validation loss.

## E.2. Decision Making with Covariance-Aware Transformers

### E.2.1. DATA GENERATION

We utilize LMC to generate synthetic financial return data with dynamic covariance structures. The generative process uses the following specific hyperparameters to simulate market conditions: `dirichlet_min=0.01`, `dirichlet_max=0.75`, `weibull_shape=6`, and `weibull_scale=125`. Each generated time series has a length of $T = 1024$. For our experiments, we construct training datasets of size $N_{\text{train}} = 300$ and evaluation datasets of size $N_{\text{eval}} = 30$. The input sequence length (lookback window) $L$ is set to 96.

### E.2.2. INFERENCE

During inference, we utilize a rolling window protocol across all strategies. At each time step $t$, the system accesses a historical context window to update the portfolio. The specific mechanism for computing the next allocation $\mathbf{s}_{t+1}$ depends on the strategy:

- **(i) Oracle**: At time $t$, it uses the *ground-truth* future return $\mathbf{r}_{t+1}$ (perfect foresight) and the empirical covariance matrix $\hat{\boldsymbol{\Sigma}}_t$ estimated from the previous $L = 96$ returns. It solves the constrained QP problem using the OSQP solver via the CVXPY interface:

$$\max_{\mathbf{s}_{t+1}} \quad \mathbf{s}_{t+1}^\top \mathbf{r}_{t+1} - \lambda \mathbf{s}_{t+1}^\top \hat{\boldsymbol{\Sigma}}_t \mathbf{s}_{t+1}$$
$$\text{s.t.} \quad \mathbf{s}_{t+1} \in \Delta_m, \quad \|\mathbf{s}_{t+1} - \mathbf{s}_t\|_1 \leq \gamma$$

where $\Delta_m$ is the standard simplex. The resulting $\mathbf{s}_{t+1}$ is utilized for the next step's turnover constraint.

- **(ii) Uniform**: A naive baseline where the portfolio is equally distributed across all assets at every step, i.e., $\mathbf{s}_{t+1,i} = \frac{1}{m}$.

- **(iii) Predict-then-Optimize**: First, a pretrained TimePFN model uses the history of actual returns $\mathbf{r}_{t-L:t}$ to forecast the expected future return vector $\hat{\mathbf{r}}_{t+1}$. The empirical covariance matrix $\hat{\mathbf{\Sigma}}_t$ is estimated from the actual history $\mathbf{r}_{t-L:t}$. We then solve the same QP problem as the Oracle, but using the *predicted* return $\hat{\mathbf{r}}_{t+1}$ instead of the ground truth.

- **(iv) End-to-End Neural Strategies (Time2Decide, Pretrained, SFT)**: These models generate allocations directly without solving a QP explicitly. At time $t$, the input window $H_t$ consists of past *actual* returns $\mathbf{r}_{t-L:t}$ and past *projected* allocations $\mathbf{s}_{t-L:t}$. The models output a raw prediction vector $\tilde{\mathbf{s}}_{t+1}$. To ensure validity, we apply a differentiable projection layer that maps $\tilde{\mathbf{s}}_{t+1}$ to the closest valid allocation $\hat{\mathbf{s}}_{t+1}$ satisfying the simplex and turnover constraints:

$$\min_{\hat{\mathbf{s}}_{t+1}} \quad \|\hat{\mathbf{s}}_{t+1} - \tilde{\mathbf{s}}_{t+1}\|_2^2$$
$$\text{s.t.} \quad \hat{\mathbf{s}}_{t+1} \in \Delta_m, \quad \|\hat{\mathbf{s}}_{t+1} - \mathbf{s}_t\|_1 \leq \gamma$$

This projection is solved using an SLSQP solver. The window for the next time step $t+1$ is updated autoregressively: we discard the oldest observation and append the *actual* realized return $\mathbf{r}_{t+1}$ and the *valid* projected allocation $\hat{\mathbf{s}}_{t+1}$ to the history.

### E.2.3. TRAINING

We employ a Supervised Fine-Tuning (SFT) approach to train both the *Time2Decide* model and the *SFT-TimePFN* baseline. Both models are initialized from a pretrained TimePFN checkpoint. The training objective is a combined loss function that jointly minimizes the error in return forecasting and portfolio allocation:

$$\mathcal{L} = \lambda_r \|\hat{\mathbf{r}} - \mathbf{r}\|_2^2 + \|\hat{\mathbf{s}} - \mathbf{s}^\star\|_2^2$$

where $\hat{\mathbf{r}}$ and $\mathbf{r}$ denote the predicted and ground truth return vectors, $\hat{\mathbf{s}}$ and $\mathbf{s}^\star$ denote the predicted and Oracle (optimal) allocation vectors, and $\lambda_r$ is a hyperparameter that weights the return loss. We conduct a comprehensive hyperparameter sweep for $\lambda_r \in \{0, 10, 50, \ldots, 320000\}$ to identify the optimal configuration for each model. Optimization is performed using the Adam optimizer with a maximum learning rate of $1 \times 10^{-4}$ and a OneCycleLR scheduler with `pct_start=0.3`. The models are trained for a maximum of 50 epochs with a batch size of 32, employing early stopping with a patience of 5 epochs to prevent overfitting.

### E.2.4. EVALUATION METRICS

The metric for experimental evaluation is the Mean Squared Error (MSE) relative to the optimal Oracle strategy.

Crucially, to rigorously assess the models' learning capability, we compute the MSE using the *raw* model predictions $\tilde{\mathbf{s}}$ (before the feasibility projection described in the Inference section) against the Oracle allocations:

$$\text{MSE} = \frac{1}{T \cdot N_{\text{eval}}} \sum_{n=1}^{N_{\text{eval}}} \sum_{t=1}^{T} \|\tilde{\mathbf{s}}_{n,t} - \mathbf{s}_{n,t}^\star\|_2^2$$

*Table 6.* $R^2$ Performance under Distribution Shift

| $\kappa$ Range | LinearTransformer | | SoftmaxTransformer | |
|---|---|---|---|---|
| | $R^2$ | Drop | $R^2$ | Drop |
| No $\kappa$ | 0.973832 | – | 0.935439 | – |
| 1.2–2 | 0.971755 | -0.21% | 0.930962 | -0.48% |
| 2–5 | 0.968110 | -0.59% | 0.924842 | -1.13% |
| 5–10 | 0.960139 | -1.41% | 0.909774 | -2.74% |
| 10–20 | 0.951526 | -2.29% | 0.888270 | -5.04% |

## F. Robustness to distribution shift

To investigate the robustness of our models, we conduct experiments where the training and test distributions differ in the condition number of the QP problems. The condition number, $\kappa(A) = \lambda_{\max}(A)/\lambda_{\min}(A)$, controls the numerical difficulty of the optimization, with higher values corresponding to more ill-conditioned problems that are harder to solve. We generate this distribution shift by controlling the eigenvalue spread of the $A$ matrix using a scaling parameter $\kappa$, which creates exponentially more difficult problems for higher $\kappa$ values. To isolate this effect, we normalize the Frobenius norm of $A$ to match the baseline distribution.

The results of this analysis are shown in Table 6. We trained the best-performing models (LinearTransformer with 8 layers/2 heads and SoftmaxTransformer with 16 layers/2 heads) on the baseline distribution and tested them on shifted distributions with varying $\kappa$ ranges. Both architectures demonstrate considerable robustness, indicating they can generalize reasonably well to more challenging numerical conditions. Again, the LinearTransformer consistently shows superior performance and a more graceful degradation across all $\kappa$ ranges as the problem difficulty increases.

The strong performance and robustness of our transformer-based QP solvers motivate their integration into foundation models for sequential decision-making. We demonstrate that QP solving can be seamlessly incorporated through supervised fine-tuning by simply adding relevant tokens to the input sequence. The following section showcases this integration through a portfolio optimization case study, where we embed QP solving within a time-series foundation model to enable end-to-end decision-making under complex structural constraints.

## G. Architectural Ablation: MLP, LSTM vs. Transformer

We included MLP and LSTM baselines designed specifically for QP inputs for comparison. The **MLP** is a deep residual network that flattens the input tokens into a single vector. It employs a 6-layer architecture with widths ranging from $6\times$ to $4\times$ the hidden dimension ($d = 256$), utilizing pre-activation LayerNorm, ReLU, dropout, and projection-based residual connections to ensure trainability. The **LSTM** baseline processes the QP tokens sequentially using a 4-layer stacked LSTM with the same hidden dimension. Crucially, the LSTM shares the exact same input embedding and output MLP head as the transformer models, ensuring that performance differences arise solely from the sequence modeling mechanism (recurrence vs. attention). To ensure a fair comparison, the parameter counts of both baselines are matched to those of our largest transformer model. Below, we present the $R^2$ results for the different models, where each problem dimension is evaluated using its best-performing transformer architecture.

*Table 7.* $R^2$ Comparison: Transformer architectures vs. MLP and LSTM baselines.

| Problem Dimensions | | Model Architecture | | | |
|---|---|---|---|---|---|
| $n$ | $m$ | LinearTransformer | SoftmaxTransformer | MLP | LSTM |
| 5 | 3 | 0.9738 | 0.9354 | 0.6715 | 0.7045 |
| 7 | 3 | 0.9600 | 0.8914 | 0.5271 | 0.4734 |
| 7 | 6 | 0.9355 | 0.8713 | 0.5071 | 0.4880 |
| 10 | 3 | 0.9054 | 0.8423 | 0.3566 | 0.2189 |
| 10 | 6 | 0.8921 | 0.8157 | 0.3642 | 0.2357 |
| 10 | 9 | 0.9062 | 0.8430 | 0.4043 | 0.2693 |

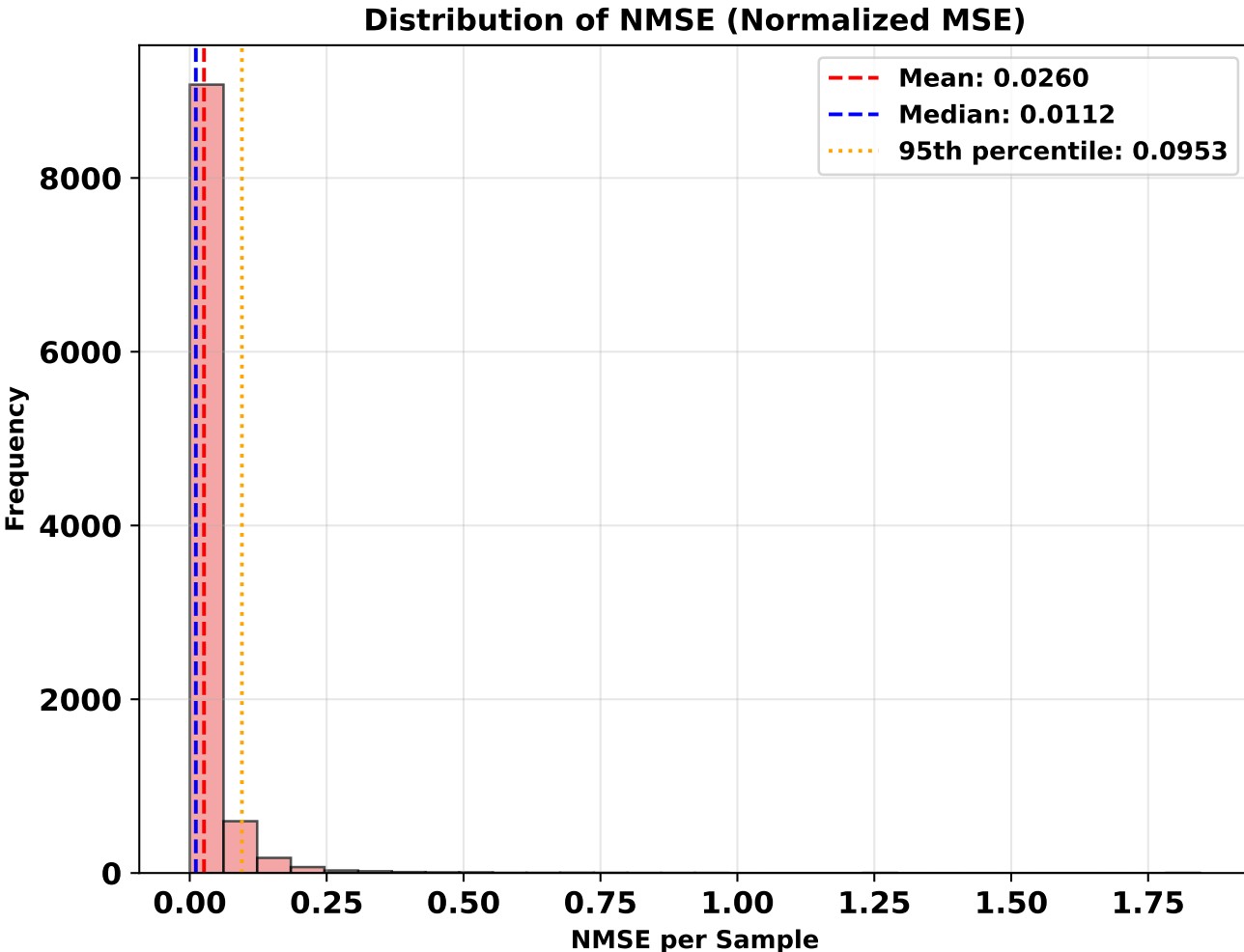

*Figure 2.* Test-set NMSE histogram for the LinearTransformer (8 layers, 2 heads). Dashed lines mark the mean and median; the dotted line marks the 95th percentile. Errors are concentrated near zero (median $\approx 0.011$; 95% $\approx 0.095$).

# H. Additional Neural QP Solver Results

We present more results for neural QP solver experiments across higher problem dimensions beyond the main text's focus on $n = 5, m = 3$ configurations. See Tables 8, 9, 10, 11, and 12 for comprehensive $R^2$ results across problem sizes $(n = 7, m = 3)$, $(n = 7, m = 6)$, $(n = 10, m = 3)$, $(n = 10, m = 6)$, and $(n = 10, m = 9)$ respectively.

These comprehensive results demonstrate that transformers are capable of solving unseen QP problems of higher dimensions. Again, LinearTransformer consistently outperforms SoftmaxTransformer across problem dimensions and hyperparameter combinations.

The entries marked as NaN indicate that the Transformer architecture with the specified depth $l$ and hidden dimension $h$ was unable to learn the QP solution for the given higher-dimensional problem and the data size, resulting in a negative $R^2$.

**Additional Error Distributions.** Figures 3 through 8 present error distribution analyzes for the best-performing configurations in higher-dimensional challenging QP problems ($n = 10$). These distributions complement performance metrics $R^2$ by providing detailed information on the error characteristics of our transformer-based solvers. The figures show NMSE distributions, constraint violation patterns, and objective error distributions, revealing how both LinearTransformer and SoftmaxTransformer architectures handle the increased complexity of larger-scale QP instances.

*Table 8. $R^2$ on the QP Function-Approximation Task ($n$=7, $m$=3).*

| Number of Layers | SoftmaxTransformer | | | | LinearTransformer | | | |
|---|---|---|---|---|---|---|---|---|
| | Number of Heads | | | | Number of Heads | | | |
| | 1 | 2 | 4 | 8 | 1 | 2 | 4 | 8 |
| 1 | 0.226 | 0.253 | 0.315 | 0.472 | 0.239 | 0.299 | 0.377 | 0.478 |
| 2 | 0.485 | 0.544 | 0.482 | 0.485 | 0.652 | 0.755 | 0.784 | 0.811 |
| 4 | 0.782 | 0.829 | 0.844 | 0.814 | 0.928 | 0.948 | 0.942 | 0.937 |
| 8 | 0.864 | 0.891 | 0.890 | 0.883 | **0.960** | 0.952 | 0.949 | 0.943 |
| 16 | 0.871 | 0.878 | 0.876 | **0.891** | 0.907 | 0.927 | 0.937 | 0.937 |

*Table 9. $R^2$ on the QP Function-Approximation Task ($n$=7, $m$=6).*

| Number of Layers | SoftmaxTransformer | | | | LinearTransformer | | | |
|---|---|---|---|---|---|---|---|---|
| | Number of Heads | | | | Number of Heads | | | |
| | 1 | 2 | 4 | 8 | 1 | 2 | 4 | 8 |
| 1 | 0.260 | 0.286 | 0.335 | 0.484 | 0.285 | 0.329 | 0.394 | 0.488 |
| 2 | 0.452 | 0.462 | 0.511 | 0.535 | 0.721 | 0.729 | 0.749 | 0.766 |
| 4 | 0.751 | 0.807 | 0.797 | 0.763 | 0.897 | 0.920 | 0.925 | 0.915 |
| 8 | 0.847 | 0.870 | 0.867 | **0.871** | 0.929 | 0.932 | **0.936** | 0.928 |
| 16 | 0.863 | 0.866 | 0.865 | 0.869 | 0.914 | 0.913 | 0.922 | 0.927 |

*Table 10.* $R^2$ on the QP Function-Approximation Task ($n=10$, $m=3$).

| Number of Layers | SoftmaxTransformer | | | | LinearTransformer | | | |
|---|---|---|---|---|---|---|---|---|
| | Number of Heads | | | | Number of Heads | | | |
| | 1 | 2 | 4 | 8 | 1 | 2 | 4 | 8 |
| 1 | 0.186 | 0.195 | 0.213 | 0.242 | 0.191 | 0.221 | 0.261 | 0.298 |
| 2 | 0.336 | 0.361 | 0.357 | 0.357 | 0.645 | 0.674 | 0.669 | 0.656 |
| 4 | 0.666 | 0.657 | 0.670 | 0.661 | 0.867 | 0.893 | 0.896 | 0.879 |
| 8 | 0.785 | 0.813 | 0.842 | 0.799 | 0.900 | **0.905** | 0.893 | 0.899 |
| 16 | 0.774 | 0.786 | 0.822 | **0.842** | 0.600 | 0.860 | 0.884 | 0.873 |

*Table 11.* $R^2$ on the QP Function-Approximation Task ($n=10$, $m=6$).

| Number of Layers | SoftmaxTransformer | | | | LinearTransformer | | | |
|---|---|---|---|---|---|---|---|---|
| | Number of Heads | | | | Number of Heads | | | |
| | 1 | 2 | 4 | 8 | 1 | 2 | 4 | 8 |
| 1 | 0.207 | 0.215 | 0.230 | 0.258 | 0.212 | 0.242 | 0.270 | 0.320 |
| 2 | 0.341 | 0.342 | 0.363 | 0.364 | 0.554 | 0.665 | 0.667 | 0.641 |
| 4 | 0.603 | 0.663 | 0.651 | 0.642 | 0.816 | 0.868 | 0.852 | 0.856 |
| 8 | 0.792 | 0.799 | 0.786 | 0.796 | **0.892** | 0.889 | 0.869 | 0.873 |
| 16 | 0.795 | 0.797 | 0.783 | **0.816** | 0.614 | 0.843 | 0.855 | 0.864 |

*Table 12.* $R^2$ on the QP Function-Approximation Task ($n=10$, $m=9$).

| Number of Layers | SoftmaxTransformer | | | | LinearTransformer | | | |
|---|---|---|---|---|---|---|---|---|
| | Number of Heads | | | | Number of Heads | | | |
| | 1 | 2 | 4 | 8 | 1 | 2 | 4 | 8 |
| 1 | 0.247 | 0.256 | 0.273 | 0.309 | 0.251 | 0.279 | 0.317 | 0.371 |
| 2 | 0.382 | 0.397 | 0.505 | 0.485 | 0.645 | 0.683 | 0.679 | 0.667 |
| 4 | 0.626 | 0.659 | 0.755 | 0.746 | 0.850 | 0.880 | 0.892 | 0.882 |
| 8 | 0.800 | 0.833 | 0.825 | 0.838 | 0.900 | **0.906** | 0.895 | 0.896 |
| 16 | 0.828 | 0.826 | 0.824 | **0.843** | 0.762 | 0.864 | 0.890 | 0.900 |

*Table 13.* $R^2$ on the QP Function-Approximation Task ($n=15$, $m=10$).

| Number of Layers | SoftmaxTransformer | | | | LinearTransformer | | | |
|---|---|---|---|---|---|---|---|---|
| | Number of Heads | | | | Number of Heads | | | |
| | 1 | 2 | 4 | 8 | 1 | 2 | 4 | 8 |
| 1 | 0.198 | 0.208 | 0.213 | 0.226 | 0.210 | 0.219 | 0.236 | 0.259 |
| 2 | 0.306 | 0.325 | 0.338 | 0.333 | 0.608 | 0.636 | 0.639 | 0.628 |
| 4 | 0.636 | 0.706 | 0.739 | 0.753 | 0.861 | 0.895 | 0.889 | 0.876 |
| 8 | 0.767 | 0.815 | **0.824** | 0.820 | 0.854 | 0.902 | **0.911** | 0.911 |

*Table 14.* $R^2$ on the QP Function-Approximation Task ($n=20$, $m=8$).

| Number of Layers | SoftmaxTransformer | | | | LinearTransformer | | | |
|---|---|---|---|---|---|---|---|---|
| | Number of Heads | | | | Number of Heads | | | |
| | 1 | 2 | 4 | 8 | 1 | 2 | 4 | 8 |
| 1 | 0.150 | 0.156 | 0.155 | 0.158 | 0.161 | 0.158 | 0.157 | 0.155 |
| 2 | 0.208 | 0.201 | 0.199 | 0.196 | 0.462 | 0.561 | 0.540 | 0.537 |
| 4 | 0.351 | 0.360 | 0.370 | 0.297 | 0.821 | 0.828 | 0.754 | 0.793 |
| 8 | 0.510 | **0.523** | 0.469 | 0.466 | NaN | **0.846** | 0.842 | 0.828 |

*Table 15.* $R^2$ on the QP Function-Approximation Task ($n{=}20$, $m{=}16$).

| Number of Layers | SoftmaxTransformer | | | | LinearTransformer | | | |
|:---:|:---:|:---:|:---:|:---:|:---:|:---:|:---:|:---:|
| | Number of Heads | | | | Number of Heads | | | |
| | 1 | 2 | 4 | 8 | 1 | 2 | 4 | 8 |
| **1** | 0.211 | 0.220 | 0.222 | 0.229 | 0.218 | 0.226 | 0.224 | 0.228 |
| **2** | 0.305 | 0.342 | 0.332 | 0.342 | 0.616 | 0.637 | 0.634 | 0.532 |
| **4** | 0.599 | 0.671 | 0.720 | 0.732 | **0.887** | 0.877 | 0.849 | 0.868 |
| **8** | 0.750 | **0.800** | 0.764 | 0.795 | NaN | NaN | 0.347 | 0.883 |

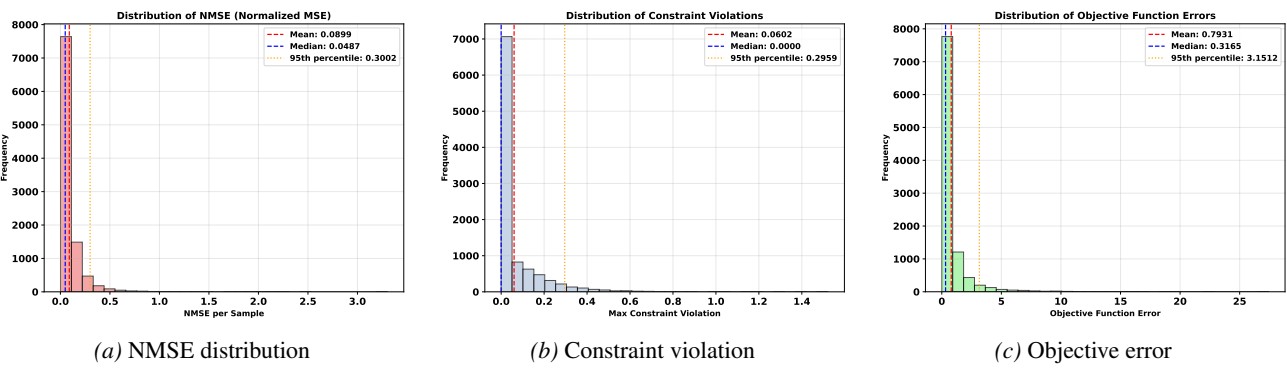

*(a)* NMSE distribution      *(b)* Constraint violation      *(c)* Objective error

*Figure 3.* Error distributions for the best LinearTransformer configuration ($n = 10, m = 3$, layers=8, heads=2).

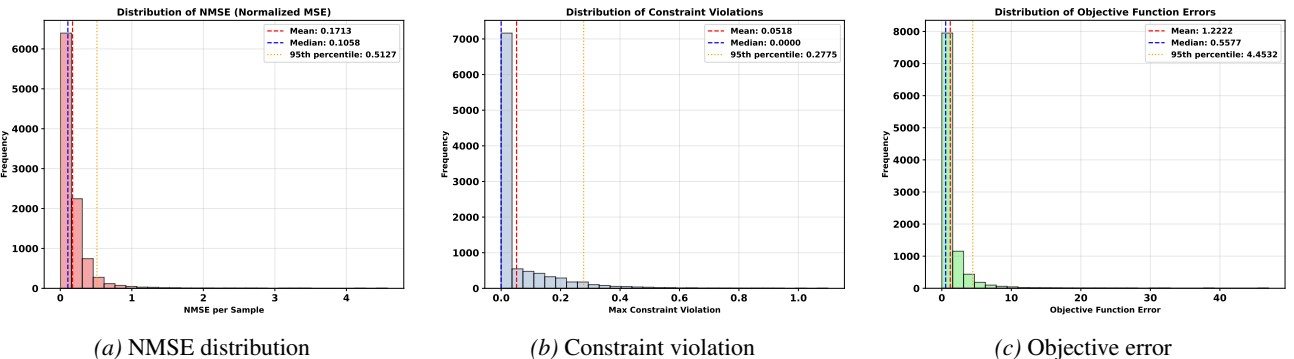

*(a)* NMSE distribution      *(b)* Constraint violation      *(c)* Objective error

*Figure 4.* Error distributions for the best SoftmaxTransformer configuration ($n = 10, m = 3$, layers=16, heads=8).

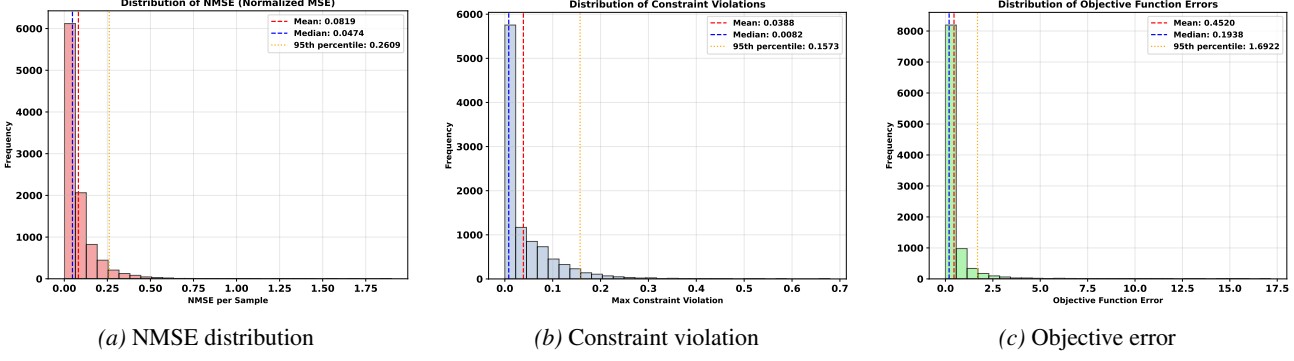

*(a)* NMSE distribution      *(b)* Constraint violation      *(c)* Objective error

*Figure 5.* Error distributions for the best LinearTransformer configuration ($n = 10, m = 6$, layers=8, heads=1).

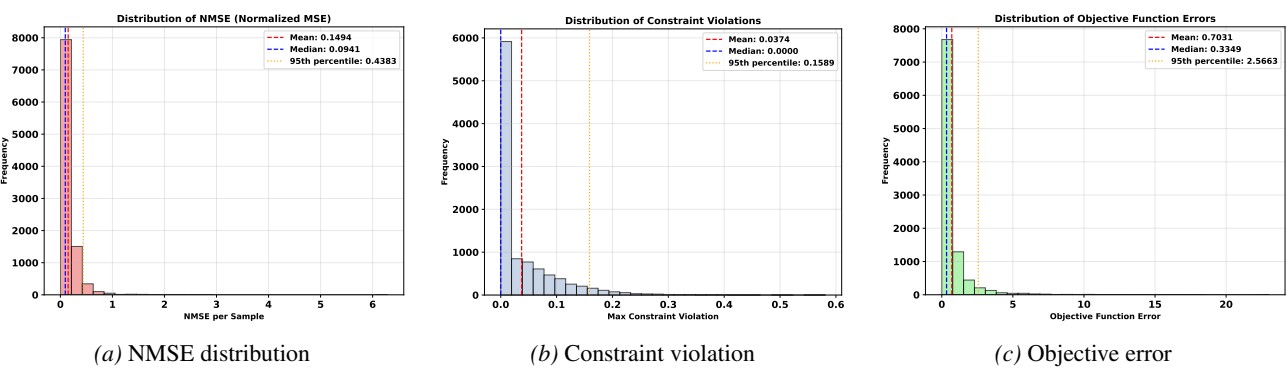

*(a)* NMSE distribution        *(b)* Constraint violation        *(c)* Objective error

*Figure 6.* Error distributions for the best SoftmaxTransformer configuration ($n = 10, m = 6$, layers=16, heads=8).

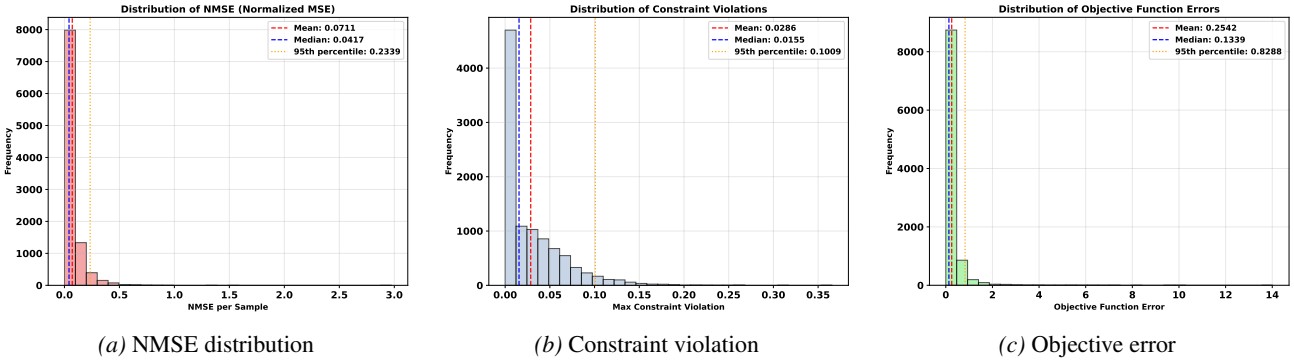

*(a)* NMSE distribution        *(b)* Constraint violation        *(c)* Objective error

*Figure 7.* Error distributions for the best LinearTransformer configuration ($n = 10, m = 9$, layers=8, heads=2).

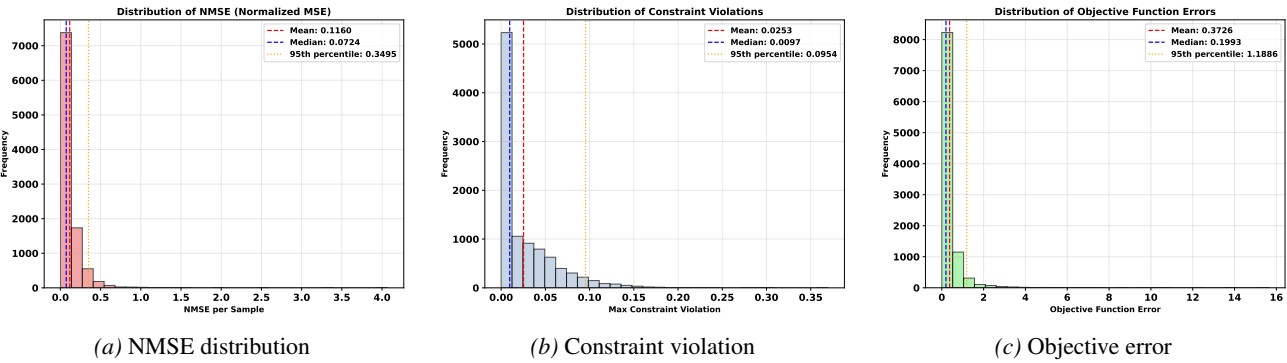

*(a)* NMSE distribution        *(b)* Constraint violation        *(c)* Objective error

*Figure 8.* Error distributions for the best SoftmaxTransformer configuration ($n = 10, m = 9$, layers=16, heads=8).

# I. Toy Model: Statistical Suboptimality of Predict-then-Optimize

**Proposition I.1** (Informal: PtO is suboptimal under estimation noise)**.** *In a two-asset mean–variance problem with Gaussian prior on the mean gap and noisy observations, the PtO plug-in rule is strictly dominated in expected utility by the Bayes-optimal policy, which shrinks the estimate by a factor $\rho < 1$ before optimizing.*

While the optimal shrinkage depends on unknown noise parameters, an end-to-end model can learn analogous attenuation directly from the decision loss. Providing covariance information helps the model calibrate how aggressively to react to forecast signals. We formalize this result in Appendix I.

We validate this insight through a portfolio optimization case study, where allocations must satisfy non-negativity, budget, and L1 rebalancing constraints. This setting highlights a practical way to integrate second-order information into foundation-model backbones for decision-making in an end-to-end fashion.

**Problem Setup.** Consider two assets with random returns $r_1, r_2$ having unknown means $\mu_1, \mu_2$. Let $w \in [0, 1]$ be the weight on asset 1. Define the mean gap $\delta := \mu_1 - \mu_2$. For simplicity, we assume the assets are uncorrelated with equal variance. Consequently, the portfolio variance is proportional to $w^2 + (1-w)^2$, i.e.,

$$\mathrm{Var}\big(wr_1 + (1-w)r_2\big) \;=\; \sigma^2\big(w^2 + (1-w)^2\big).$$

Without loss of generality, we absorb the scale factor $\sigma^2$ into the risk-aversion parameter by redefining $\lambda \leftarrow \lambda\sigma^2$; thus $\lambda$ below should be interpreted as the coefficient on the variance term. We adopt a standard mean-variance utility:

$$U(w;\delta) \;:=\; \mu_2 + w\delta - \lambda\big(w^2 + (1-w)^2\big), \quad \lambda > 0.$$

By completing the square for the quadratic penalty, maximizing the utility is equivalent to minimizing the distance between $w$ and a shifted target. Letting $u := w - 1/2$, the objective simplifies to:

$$\max_{u \in [-1/2, 1/2]} \quad u\delta - 2\lambda u^2.$$

**Assumptions.** This toy model makes several simplifying assumptions to enable closed-form insights. First, we consider only two assets with uncorrelated returns and equal variances, so the portfolio variance simplifies to $\sigma^2\big(w^2 + (1-w)^2\big)$. Second, we use a mean–variance objective with fixed risk aversion $\lambda > 0$ and a box constraint $w \in [0, 1]$, which induces the clipped linear decision rule. Third, we adopt a Bayesian formulation in which the true mean gap $\delta$ follows a Gaussian prior $\mathcal{N}(0, \tau^2)$ and the estimate $\hat{\delta}$ is a noisy Gaussian signal; this conjugate structure yields the posterior mean $\mathbb{E}[\delta \mid \hat{\delta}]$ and the corresponding linear shrinkage factor in closed form.

**Oracle vs. Predict-then-Optimize.** If the true gap $\delta$ were known, the optimal allocation $w^\star$ (derived by setting the derivative to zero and clipping to the feasible set) would be:

$$w^\star(\delta) \;=\; \Pi_{[0,1]}\left(\frac{1}{2} + \frac{\delta}{4\lambda}\right).$$

In the standard **Predict-then-Optimize (PtO)** approach, we observe samples, compute the empirical mean gap $\hat{\delta} = \hat{\mu}_1 - \hat{\mu}_2$, and simply plug it into the oracle formula:

$$w_{\mathrm{pto}}(\hat{\delta}) \;=\; \Pi_{[0,1]}\left(\frac{1}{2} + \frac{\hat{\delta}}{4\lambda}\right).$$

**End-to-End Learning.** In contrast, an end-to-end learner seeks the action that maximizes expected utility *given* the noisy signal $\hat{\delta}$. This corresponds to maximizing the conditional expectation:

$$\mathbb{E}[U(w;\delta) \mid \hat{\delta}] \;=\; \mathrm{const} + u\,\mathbb{E}[\delta \mid \hat{\delta}] - 2\lambda u^2.$$

The solution to this problem uses the *posterior mean* of the gap, not the raw estimate:

$$w_{\mathrm{e2e}}(\hat{\delta}) \;=\; \Pi_{[0,1]}\left(\frac{1}{2} + \frac{\mathbb{E}[\delta|\hat{\delta}]}{4\lambda}\right).$$

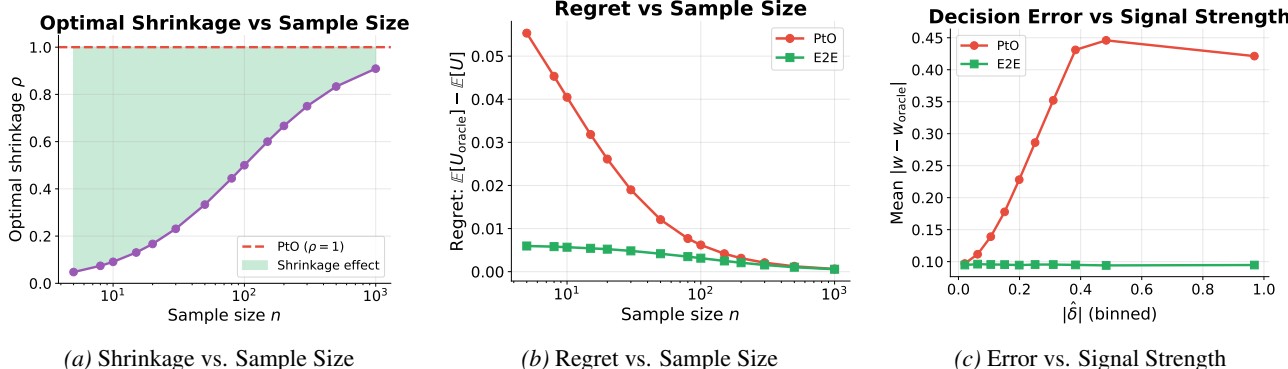

*(a)* Shrinkage vs. Sample Size      *(b)* Regret vs. Sample Size      *(c)* Error vs. Signal Strength

*Figure 9.* Empirical verification of predict-then-optimize (PtO) suboptimality compared to end-to-end (E2E) learning. **(a)** The optimal shrinkage factor $\rho = \tau^2/(\tau^2 + s^2)$ increases toward 1 as sample size $n$ grows; the shaded region shows the shrinkage effect that E2E exploits while PtO uses $\rho = 1$. **(b)** Regret for both methods; E2E consistently achieves lower regret than PtO across all sample sizes. **(c)** Decision error $|w - w_{\mathrm{oracle}}|$ as a function of observed signal strength $|\hat{\delta}|$; PtO error grows linearly with signal strength while E2E remains stable due to shrinkage. Parameters: $\tau^2 = 0.01$, $\sigma^2 = 0.5$, $\lambda = 0.2$.

**The Gaussian Case: Explicit Shrinkage.** To see why the end-to-end policy outperforms PtO, assume a Gaussian prior on the true gap $\delta \sim \mathcal{N}(0, \tau^2)$ and a sampling distribution $\hat{\delta} \sim \mathcal{N}(\delta, s^2)$ (where $s^2 \propto 1/n$). In particular, under the i.i.d. Gaussian return model with $n$ samples per asset, we have

$$s^2 \;=\; \frac{2\sigma^2}{n}.$$

The posterior expectation is a linear shrinkage of the observed signal:

$$\mathbb{E}[\delta|\hat{\delta}] = \rho\hat{\delta}, \quad \text{where } \rho = \frac{\tau^2}{\tau^2 + s^2} \in (0, 1).$$

Substituting this into the general end-to-end update yields the explicit decision rule:

$$w_{\mathrm{e2e}}(\hat{\delta}) \;=\; \Pi_{[0,1]}\left(\frac{1}{2} + \rho\frac{\hat{\delta}}{4\lambda}\right).$$

Comparing the equations for $w_{\mathrm{pto}}$ and $w_{\mathrm{e2e}}$ reveals that PtO implicitly assumes $\rho = 1$. However, because $\rho < 1$ in finite samples, the end-to-end model outperforms PtO by **shrinking** the noisy estimate $\hat{\delta}$ toward the prior (zero), thereby reducing the variance of the decision. Moreover, since $w_{\mathrm{e2e}}(\hat{\delta})$ maximizes the conditional expected utility for each fixed $\hat{\delta}$, for any measurable policy $w(\hat{\delta}) \in [0, 1]$ we have

$$\mathbb{E}\left[U\left(w_{\mathrm{e2e}}(\hat{\delta}); \delta\right)\Big|\hat{\delta}\right] \;\geq\; \mathbb{E}\left[U\left(w(\hat{\delta}); \delta\right)\Big|\hat{\delta}\right].$$

Taking expectations over $\hat{\delta}$ and using the tower property yields

$$\mathbb{E}\left[U\left(w_{\mathrm{e2e}}(\hat{\delta}); \delta\right)\right] \;\geq\; \mathbb{E}\left[U\left(w(\hat{\delta}); \delta\right)\right].$$

In particular, choosing $w(\hat{\delta}) = w_{\mathrm{pto}}(\hat{\delta})$ gives

$$\mathbb{E}\left[U\left(w_{\mathrm{e2e}}(\hat{\delta}); \delta\right)\right] \;\geq\; \mathbb{E}\left[U\left(w_{\mathrm{pto}}(\hat{\delta}); \delta\right)\right].$$

**Empirical verification.** We empirically validate the conclusions of the toy model by Monte Carlo simulation. In each trial, we sample $\delta \sim \mathcal{N}(0, \tau^2)$ and generate a noisy estimate $\hat{\delta} \mid \delta \sim \mathcal{N}(\delta, s^2)$ with $s^2 = 2\sigma^2/n$. We then compute the PtO and end-to-end decisions

$$w_{\mathrm{pto}}(\hat{\delta}) = \Pi_{[0,1]}\left(\frac{1}{2} + \frac{\hat{\delta}}{4\lambda}\right), \qquad w_{\mathrm{e2e}}(\hat{\delta}) = \Pi_{[0,1]}\left(\frac{1}{2} + \rho\frac{\hat{\delta}}{4\lambda}\right), \qquad \rho = \frac{\tau^2}{\tau^2 + s^2},$$

and evaluate the realized utility

$$U(w;\delta) = \mu_2 + w\delta - \lambda\big(w^2 + (1-w)^2\big),$$

averaging over $10^5$ trials to estimate expected utilities. We also compute the oracle decision $w_{\text{oracle}}(\delta) = \Pi_{[0,1]}\big(\frac{1}{2} + \frac{\delta}{4\lambda}\big)$, which has access to the true $\delta$, and define regret as the expected utility loss relative to the oracle: $\text{Regret}(w) = \mathbb{E}[U(w_{\text{oracle}})] - \mathbb{E}[U(w)]$.

**Results.** Figure 9 reports three key findings. Panel (a) shows the optimal shrinkage factor $\rho$ as a function of sample size $n$; the shaded region represents the shrinkage effect that E2E exploits while PtO implicitly uses $\rho = 1$. Panel (b) compares the regret of both methods: E2E consistently achieves lower regret than PtO across all sample sizes, with the gap largest in the low-data regime and vanishing as $n \to \infty$ (where $\rho \to 1$). Panel (c) analyzes decision error $|w - w_{\text{oracle}}|$ as a function of observed signal strength $|\hat{\delta}|$: PtO error grows linearly with $|\hat{\delta}|$ because it overreacts to noisy estimates, while E2E error remains stable due to shrinkage. These results confirm the theoretical prediction that PtO is statistically suboptimal under finite-sample estimation noise, and that the suboptimality arises precisely from the failure to shrink predictions toward the prior mean.

**Connection to Neural End-to-End Learning.** The closed-form policy

$$w_{\text{e2e}}(\hat{\delta}) = \Pi_{[0,1]}\left(\frac{1}{2} + \rho\,\frac{\hat{\delta}}{4\lambda}\right), \qquad \rho = \frac{\tau^2}{\tau^2 + s^2},$$

illustrates that a decision-aware learner should *shrink* noisy estimates before optimizing. Of course, the Bayes-optimal shrinkage factor $\rho$ depends on unknown quantities such as the prior variance $\tau^2$ and the effective sampling noise $s^2$, so it cannot be hard-coded in practice. However, a neural model trained end-to-end on historical data can *implicitly learn* an analogous attenuation of noisy signals: by optimizing a downstream decision objective (e.g., utility/regret or supervision from oracle allocations), gradient-based training favors mappings from inputs to actions that are robust to estimation uncertainty. This perspective helps explain why TIME2DECIDE can outperform a Predict–Optimize baseline in our experiments. Whereas PtO effectively corresponds to using $\rho = 1$, TIME2DECIDE can learn to temper its responses when the signal-to-noise ratio is low (e.g., $\text{SNR} := |\hat{\delta}|/s$, where $s^2 = 2\sigma^2/n$ in the toy model). In particular, providing covariance information via our CAT tokenization gives the model explicit second-order statistics about the market environment, which can inform *how much* to trust the return signal and yield more calibrated decisions under uncertainty.

## J. Negativity Case: Parity Barrier for Pure-Data Linear Attention

We show a simple obstruction: with only pure data tokens, linear attention cannot synthesize even-degree statistics (e.g., covariance). Intuitively, every attention layer contributes one factor of the data, so any finite stack remains an odd-degree function of the inputs unless we inject constants or explicit second-order tokens.

**Proposition J.1.** *Let tokens be $z_i = x_i \in \mathbb{R}^n$ for $i = 1,\dots,N$ and let all per-token maps be linear: $q_j = x_j W_Q$, $k_i = x_i W_K$, $v_i = x_i W_V$. With linear attention $o_j = \sum_{i=1}^N \langle q_j, k_i \rangle v_i$, we have*

$$o_j = \Big[\sum_{i=1}^N (Vx_i)x_i^\top\Big] M^\top x_j, \qquad M := W_Q W_K^\top, \ V := W_V,$$

*so $o_j$ is linear in $x_j$ and quadratic in $\{x_i\}_{i=1}^N$, hence an* odd-*degree function of the dataset. Consequently, such a layer and any finite stack cannot construct even-degree statistics like $\sum_{i=1}^N x_i x_i^\top$ or any map invariant under the global flip $x_i \mapsto -x_i$.*

*Proof.* We compute

$$\langle q_j, k_i \rangle = (x_j W_Q)(x_i W_K)^\top = x_j^\top W_Q W_K^\top x_i = x_j^\top M x_i.$$

Thus

$$o_j = \sum_{i=1}^N (x_j^\top M x_i)\,(Vx_i) = \Big(\sum_{i=1}^N (Vx_i)x_i^\top\Big) M^\top x_j =: S_V\, M^\top x_j,$$

where $S_V := \sum_{i=1}^{N}(Vx_i)x_i^\top$ is a quadratic sketch of the data. Under the global sign flip $x_i \mapsto -x_i$ for all $i = 1, \ldots, N$, we have $S_V \mapsto S_V$, while $x_j \mapsto -x_j$, so $o_j = S_V M^\top x_j \mapsto -o_j$. Thus, the map is odd and cannot equal a flip-invariant even statistic such as $\sum_{i=1}^{N} x_i x_i^\top$. Each additional linear-attention layer multiplies by one more linear factor in the data, raising the degree by one, while linear/ReLU maps do not introduce even-degree terms. Therefore, no finite-depth stack of such components can yield even-degree statistics without adding constant channels or explicit second-order tokens. □

**Empirical validation.** To validate this theoretical limitation, we conduct a simple covariance learning experiment where models must learn to estimate covariance matrices from unseen multivariate time series. We compare two transformer variants: (i) LinearTransformer with linear attention and input projection layers, and (ii) BasicLinearTransformer with linear attention and no input projection layers. Each model processes sequences of length 50 with 4 variables and predicts the empirical covariance matrix using Frobenius norm loss. The models are trained on 5000 sample series and tested on 1000 sample series.

Tables 16 and 17 summarize the R² performance across different architectures and configurations. The results confirm our theoretical prediction: LinearTransformer achieves R² 0.9935, while BasicLinearTransformer (without input projection) performs significantly worse with R² 0.2827.

| Model | Layers | Heads | Hidden | R² |
|---|---|---|---|---|
| LinearTransformer | 4 | 2 | 16 | 0.9935 |
| LinearTransformer | 4 | 4 | 16 | 0.9892 |

*Table 16.* LinearTransformer covariance learning performance (R²) with input projection layers.

| Model | Layers | Heads | R² |
|---|---|---|---|
| BasicLinearTransformer | 4 | 4 | 0.2827 |
| BasicLinearTransformer | 8 | 4 | 0.2473 |
| BasicLinearTransformer | 4 | 2 | 0.2234 |
| BasicLinearTransformer | 8 | 2 | 0.1953 |

*Table 17.* BasicLinearTransformer covariance learning performance (R²) without input projection layers.

These empirical results support our theoretical analysis: while linear attention can learn covariance patterns when equipped with proper input projections/embeddings, removing these projections creates a fundamental barrier to learning even-degree statistics.

## K. Additional Portfolio Optimization Results

This section presents comprehensive MSE results for portfolio optimization across different rebalancing constraints $\gamma \in \{0.5, 0.75, 1.0, 1.25, 1.5, 1.75, 2.0\}$ and return loss weights $\lambda_r \in \{0, 10, 50, 100, 200, 300, 500, 1000, 3000, 5000, 10000, 20000, 40000, 80000, 160000, 320000, 640000\}$. In most of our experiments, SFT-TimePFN achieves a better performance with a small $\lambda_r$ (focusing primarily on allocation prediction), while Time2Decide requires a larger $\lambda_r$ value to balance both return and allocation predictions effectively.

*Table 18.* MSE results for $\gamma = 0.5$ across different return loss weights (Part 1).

| Strategy | $\lambda_r=0$ | $\lambda_r=10$ | $\lambda_r=50$ | $\lambda_r=100$ | $\lambda_r=200$ | $\lambda_r=300$ | $\lambda_r=500$ | $\lambda_r=1000$ | $\lambda_r=3000$ |
|---|---|---|---|---|---|---|---|---|---|
| Predict-then-Optimize | 0.0245 | 0.0245 | 0.0245 | 0.0245 | 0.0245 | 0.0245 | 0.0245 | 0.0245 | 0.0245 |
| Pretrained-TimePFN | 0.0447 | 0.0447 | 0.0447 | 0.0447 | 0.0447 | 0.0447 | 0.0447 | 0.0447 | 0.0447 |
| Uniform | 0.0425 | 0.0425 | 0.0425 | 0.0425 | 0.0425 | 0.0425 | 0.0425 | 0.0425 | 0.0425 |
| SFT-TimePFN | 0.0408 | 0.0462 | 0.0559 | 0.0566 | 0.0538 | 0.0626 | 0.0610 | 0.0626 | 0.0624 |
| Time2Decide | 0.0635 | 0.0618 | 0.0521 | 0.0526 | 0.0482 | 0.0477 | 0.0439 | 0.0367 | 0.0346 |

*Table 19.* MSE results for $\gamma = 0.5$ across different return loss weights (Part 2).

| Strategy | $\lambda_r$=5000 | $\lambda_r$=10000 | $\lambda_r$=20000 | $\lambda_r$=40000 | $\lambda_r$=80000 | $\lambda_r$=160000 | $\lambda_r$=320000 | $\lambda_r$=640000 |
|---|---|---|---|---|---|---|---|---|
| Predict-then-Optimize | 0.0245 | 0.0245 | 0.0245 | 0.0245 | 0.0245 | 0.0245 | 0.0245 | 0.0245 |
| Pretrained-TimePFN | 0.0447 | 0.0447 | 0.0447 | 0.0447 | 0.0447 | 0.0447 | 0.0447 | 0.0447 |
| Uniform | 0.0425 | 0.0425 | 0.0425 | 0.0425 | 0.0425 | 0.0425 | 0.0425 | 0.0425 |
| SFT-TimePFN | 0.0622 | 0.0621 | 0.0627 | 0.0621 | 0.0610 | 0.0618 | 0.0614 | 0.0614 |
| Time2Decide | 0.0340 | 0.0339 | 0.0326 | 0.0325 | 0.0325 | 0.0323 | 0.0323 | 0.0323 |

*Table 20.* MSE results for $\gamma = 0.75$ across different return loss weights (Part 1).

| Strategy | $\lambda_r$=0 | $\lambda_r$=10 | $\lambda_r$=50 | $\lambda_r$=100 | $\lambda_r$=200 | $\lambda_r$=300 | $\lambda_r$=500 | $\lambda_r$=1000 | $\lambda_r$=3000 |
|---|---|---|---|---|---|---|---|---|---|
| Predict-then-Optimize | 0.0294 | 0.0294 | 0.0294 | 0.0294 | 0.0294 | 0.0294 | 0.0294 | 0.0294 | 0.0294 |
| Pretrained-TimePFN | 0.0458 | 0.0458 | 0.0458 | 0.0458 | 0.0458 | 0.0458 | 0.0458 | 0.0458 | 0.0458 |
| Uniform | 0.0444 | 0.0444 | 0.0444 | 0.0444 | 0.0444 | 0.0444 | 0.0444 | 0.0444 | 0.0444 |
| SFT-TimePFN | 0.0411 | 0.0499 | 0.0554 | 0.0548 | 0.0527 | 0.0635 | 0.0643 | 0.0648 | 0.0639 |
| Time2Decide | 0.0646 | 0.0607 | 0.0501 | 0.0495 | 0.0510 | 0.0476 | 0.0417 | 0.0378 | 0.0359 |

*Table 21.* MSE results for $\gamma = 0.75$ across different return loss weights (Part 2).

| Strategy | $\lambda_r$=5000 | $\lambda_r$=10000 | $\lambda_r$=20000 | $\lambda_r$=40000 | $\lambda_r$=80000 | $\lambda_r$=160000 | $\lambda_r$=320000 | $\lambda_r$=640000 |
|---|---|---|---|---|---|---|---|---|
| Predict-then-Optimize | 0.0294 | 0.0294 | 0.0294 | 0.0294 | 0.0294 | 0.0294 | 0.0294 | 0.0294 |
| Pretrained-TimePFN | 0.0458 | 0.0458 | 0.0458 | 0.0458 | 0.0458 | 0.0458 | 0.0458 | 0.0458 |
| Uniform | 0.0444 | 0.0444 | 0.0444 | 0.0444 | 0.0444 | 0.0444 | 0.0444 | 0.0444 |
| SFT-TimePFN | 0.0637 | 0.0632 | 0.0635 | 0.0634 | 0.0633 | 0.0626 | 0.0624 | 0.0637 |
| Time2Decide | 0.0352 | 0.0351 | 0.0350 | 0.0348 | 0.0348 | 0.0347 | 0.0346 | 0.0347 |

*Table 22.* MSE results for $\gamma = 1.0$ across different return loss weights (Part 1).

| Strategy | $\lambda_r$=0 | $\lambda_r$=10 | $\lambda_r$=50 | $\lambda_r$=100 | $\lambda_r$=200 | $\lambda_r$=300 | $\lambda_r$=500 | $\lambda_r$=1000 | $\lambda_r$=3000 |
|---|---|---|---|---|---|---|---|---|---|
| Predict-then-Optimize | 0.0360 | 0.0360 | 0.0360 | 0.0360 | 0.0360 | 0.0360 | 0.0360 | 0.0360 | 0.0360 |
| Pretrained-TimePFN | 0.0500 | 0.0500 | 0.0500 | 0.0500 | 0.0500 | 0.0500 | 0.0500 | 0.0500 | 0.0500 |
| Uniform | 0.0479 | 0.0479 | 0.0479 | 0.0479 | 0.0479 | 0.0479 | 0.0479 | 0.0479 | 0.0479 |
| SFT-TimePFN | 0.0413 | 0.0504 | 0.0542 | 0.0553 | 0.0528 | 0.0681 | 0.0688 | 0.0690 | 0.0680 |
| Time2Decide | 0.0629 | 0.0561 | 0.0504 | 0.0566 | 0.0503 | 0.0478 | 0.0436 | 0.0414 | 0.0387 |

*Table 23.* MSE results for $\gamma = 1.0$ across different return loss weights (Part 2).

| Strategy | $\lambda_r$=5000 | $\lambda_r$=10000 | $\lambda_r$=20000 | $\lambda_r$=40000 | $\lambda_r$=80000 | $\lambda_r$=160000 | $\lambda_r$=320000 | $\lambda_r$=640000 |
|---|---|---|---|---|---|---|---|---|
| Predict-then-Optimize | 0.0360 | 0.0360 | 0.0360 | 0.0360 | 0.0360 | 0.0360 | 0.0360 | 0.0360 |
| Pretrained-TimePFN | 0.0500 | 0.0500 | 0.0500 | 0.0500 | 0.0500 | 0.0500 | 0.0500 | 0.0500 |
| Uniform | 0.0479 | 0.0479 | 0.0479 | 0.0479 | 0.0479 | 0.0479 | 0.0479 | 0.0479 |
| SFT-TimePFN | 0.0683 | 0.0681 | 0.0669 | 0.0680 | 0.0675 | 0.0672 | 0.0673 | 0.0674 |
| Time2Decide | 0.0386 | 0.0391 | 0.0387 | 0.0387 | 0.0388 | 0.0387 | 0.0387 | 0.0386 |

*Table 24.* MSE results for $\gamma = 1.25$ across different return loss weights (Part 1).

| Strategy | $\lambda_r$=0 | $\lambda_r$=10 | $\lambda_r$=50 | $\lambda_r$=100 | $\lambda_r$=200 | $\lambda_r$=300 | $\lambda_r$=500 | $\lambda_r$=1000 | $\lambda_r$=3000 |
|---|---|---|---|---|---|---|---|---|---|
| Predict-then-Optimize | 0.0376 | 0.0376 | 0.0376 | 0.0376 | 0.0376 | 0.0376 | 0.0376 | 0.0376 | 0.0376 |
| Pretrained-TimePFN | 0.0514 | 0.0514 | 0.0514 | 0.0514 | 0.0514 | 0.0514 | 0.0514 | 0.0514 | 0.0514 |
| Uniform | 0.0485 | 0.0485 | 0.0485 | 0.0485 | 0.0485 | 0.0485 | 0.0485 | 0.0485 | 0.0485 |
| SFT-TimePFN | 0.0446 | 0.0560 | 0.0550 | 0.0523 | 0.0523 | 0.0683 | 0.0678 | 0.0685 | 0.0672 |
| Time2Decide | 0.0639 | 0.0501 | 0.0489 | 0.0498 | 0.0486 | 0.0452 | 0.0416 | 0.0400 | 0.0390 |

*Table 25.* MSE results for $\gamma = 1.25$ across different return loss weights (Part 2).

| Strategy | $\lambda_r$=5000 | $\lambda_r$=10000 | $\lambda_r$=20000 | $\lambda_r$=40000 | $\lambda_r$=80000 | $\lambda_r$=160000 | $\lambda_r$=320000 | $\lambda_r$=640000 |
|---|---|---|---|---|---|---|---|---|
| Predict-then-Optimize | 0.0376 | 0.0376 | 0.0376 | 0.0376 | 0.0376 | 0.0376 | 0.0376 | 0.0376 |
| Pretrained-TimePFN | 0.0514 | 0.0514 | 0.0514 | 0.0514 | 0.0514 | 0.0514 | 0.0514 | 0.0514 |
| Uniform | 0.0485 | 0.0485 | 0.0485 | 0.0485 | 0.0485 | 0.0485 | 0.0485 | 0.0485 |
| SFT-TimePFN | 0.0667 | 0.0681 | 0.0678 | 0.0675 | 0.0679 | 0.0678 | 0.0676 | 0.0681 |
| Time2Decide | 0.0391 | 0.0393 | 0.0394 | 0.0395 | 0.0395 | 0.0395 | 0.0395 | 0.0396 |

*Table 26.* MSE results for $\gamma = 1.5$ across different return loss weights (Part 1).

| Strategy | $\lambda_r$=0 | $\lambda_r$=10 | $\lambda_r$=50 | $\lambda_r$=100 | $\lambda_r$=200 | $\lambda_r$=300 | $\lambda_r$=500 | $\lambda_r$=1000 | $\lambda_r$=3000 |
|---|---|---|---|---|---|---|---|---|---|
| Predict-then-Optimize | 0.0413 | 0.0413 | 0.0413 | 0.0413 | 0.0413 | 0.0413 | 0.0413 | 0.0413 | 0.0413 |
| Pretrained-TimePFN | 0.0523 | 0.0523 | 0.0523 | 0.0523 | 0.0523 | 0.0523 | 0.0523 | 0.0523 | 0.0523 |
| Uniform | 0.0505 | 0.0505 | 0.0505 | 0.0505 | 0.0505 | 0.0505 | 0.0505 | 0.0505 | 0.0505 |
| SFT-TimePFN | 0.0445 | 0.0557 | 0.0529 | 0.0532 | 0.0689 | 0.0698 | 0.0693 | 0.0692 | 0.0698 |
| Time2Decide | 0.0574 | 0.0490 | 0.0472 | 0.0457 | 0.0461 | 0.0435 | 0.0423 | 0.0414 | 0.0410 |

*Table 27.* MSE results for $\gamma = 1.5$ across different return loss weights (Part 2).

| Strategy | $\lambda_r$=5000 | $\lambda_r$=10000 | $\lambda_r$=20000 | $\lambda_r$=40000 | $\lambda_r$=80000 | $\lambda_r$=160000 | $\lambda_r$=320000 | $\lambda_r$=640000 |
|---|---|---|---|---|---|---|---|---|
| Predict-then-Optimize | 0.0413 | 0.0413 | 0.0413 | 0.0413 | 0.0413 | 0.0413 | 0.0413 | 0.0413 |
| Pretrained-TimePFN | 0.0523 | 0.0523 | 0.0523 | 0.0523 | 0.0523 | 0.0523 | 0.0523 | 0.0523 |
| Uniform | 0.0505 | 0.0505 | 0.0505 | 0.0505 | 0.0505 | 0.0505 | 0.0505 | 0.0505 |
| SFT-TimePFN | 0.0683 | 0.0686 | 0.0690 | 0.0679 | 0.0677 | 0.0680 | 0.0681 | 0.0681 |
| Time2Decide | 0.0412 | 0.0415 | 0.0418 | 0.0420 | 0.0420 | 0.0420 | 0.0419 | 0.0420 |

*Table 28.* MSE results for $\gamma = 1.75$ across different return loss weights (Part 1).

| Strategy | $\lambda_r$=0 | $\lambda_r$=10 | $\lambda_r$=50 | $\lambda_r$=100 | $\lambda_r$=200 | $\lambda_r$=300 | $\lambda_r$=500 | $\lambda_r$=1000 | $\lambda_r$=3000 |
|---|---|---|---|---|---|---|---|---|---|
| Predict-then-Optimize | 0.0472 | 0.0472 | 0.0472 | 0.0472 | 0.0472 | 0.0472 | 0.0472 | 0.0472 | 0.0472 |
| Pretrained-TimePFN | 0.0567 | 0.0567 | 0.0567 | 0.0567 | 0.0567 | 0.0567 | 0.0567 | 0.0567 | 0.0567 |
| Uniform | 0.0539 | 0.0539 | 0.0539 | 0.0539 | 0.0539 | 0.0539 | 0.0539 | 0.0539 | 0.0539 |
| SFT-TimePFN | 0.0583 | 0.0515 | 0.0544 | 0.0541 | 0.0732 | 0.0724 | 0.0723 | 0.0727 | 0.0721 |
| Time2Decide | 0.0626 | 0.0538 | 0.0486 | 0.0474 | 0.0472 | 0.0457 | 0.0448 | 0.0445 | 0.0442 |

*Table 29.* MSE results for $\gamma = 1.75$ across different return loss weights (Part 2).

| Strategy | $\lambda_r$=5000 | $\lambda_r$=10000 | $\lambda_r$=20000 | $\lambda_r$=40000 | $\lambda_r$=80000 | $\lambda_r$=160000 | $\lambda_r$=320000 | $\lambda_r$=640000 |
|---|---|---|---|---|---|---|---|---|
| Predict-then-Optimize | 0.0472 | 0.0472 | 0.0472 | 0.0472 | 0.0472 | 0.0472 | 0.0472 | 0.0472 |
| Pretrained-TimePFN | 0.0567 | 0.0567 | 0.0567 | 0.0567 | 0.0567 | 0.0567 | 0.0567 | 0.0567 |
| Uniform | 0.0539 | 0.0539 | 0.0539 | 0.0539 | 0.0539 | 0.0539 | 0.0539 | 0.0539 |
| SFT-TimePFN | 0.0711 | 0.0716 | 0.0718 | 0.0720 | 0.0721 | 0.0725 | 0.0726 | 0.0722 |
| Time2Decide | 0.0444 | 0.0448 | 0.0452 | 0.0453 | 0.0454 | 0.0455 | 0.0455 | 0.0455 |

*Table 30.* MSE results for $\gamma = 2.0$ across different return loss weights (Part 1).

| Strategy | $\lambda_r$=0 | $\lambda_r$=10 | $\lambda_r$=50 | $\lambda_r$=100 | $\lambda_r$=200 | $\lambda_r$=300 | $\lambda_r$=500 | $\lambda_r$=1000 | $\lambda_r$=3000 |
|---|---|---|---|---|---|---|---|---|---|
| Predict-then-Optimize | 0.0553 | 0.0553 | 0.0553 | 0.0553 | 0.0553 | 0.0553 | 0.0553 | 0.0553 | 0.0553 |
| Pretrained-TimePFN | 0.0604 | 0.0604 | 0.0604 | 0.0604 | 0.0604 | 0.0604 | 0.0604 | 0.0604 | 0.0604 |
| Uniform | 0.0586 | 0.0586 | 0.0586 | 0.0586 | 0.0586 | 0.0586 | 0.0586 | 0.0586 | 0.0586 |
| SFT-TimePFN | 0.0620 | 0.0620 | 0.0567 | 0.0565 | 0.0769 | 0.0754 | 0.0771 | 0.0770 | 0.0771 |
| Time2Decide | 0.0618 | 0.0575 | 0.0524 | 0.0526 | 0.0506 | 0.0499 | 0.0490 | 0.0488 | 0.0488 |

*Table 31.* MSE results for $\gamma = 2.0$ across different return loss weights (Part 2).

| Strategy | $\lambda_r$=5000 | $\lambda_r$=10000 | $\lambda_r$=20000 | $\lambda_r$=40000 | $\lambda_r$=80000 | $\lambda_r$=160000 | $\lambda_r$=320000 | $\lambda_r$=640000 |
|---|---|---|---|---|---|---|---|---|
| Predict-then-Optimize | 0.0553 | 0.0553 | 0.0553 | 0.0553 | 0.0553 | 0.0553 | 0.0553 | 0.0553 |
| Pretrained-TimePFN | 0.0604 | 0.0604 | 0.0604 | 0.0604 | 0.0604 | 0.0604 | 0.0604 | 0.0604 |
| Uniform | 0.0586 | 0.0586 | 0.0586 | 0.0586 | 0.0586 | 0.0586 | 0.0586 | 0.0586 |
| SFT-TimePFN | 0.0766 | 0.0768 | 0.0769 | 0.0767 | 0.0763 | 0.0772 | 0.0771 | 0.0770 |
| Time2Decide | 0.0491 | 0.0503 | 0.0502 | 0.0506 | 0.0506 | 0.0505 | 0.0507 | 0.0506 |

