# OpenReview forum: "Covariance-Aware Transformers for Quadratic Programming and Decision Making"
_ICML.cc/2026/Workshop/FMSD — FMSD @ ICML 2026 Poster_

### Official Review · Reviewer_xA5w · 2026-05-18

**Rating:** 5
**Confidence:** 4

**Review:**

## Summary

The paper introduces transformer’s ability to solve quadratic programming, and how it utilises such ability to handle portfolio optimisation.
Specifically, authors empirically verify that transformer, when given the matrix and vectors in the problem as sequence, transformer can predict the solution with high accuracy, and theoretically, show that linear attention layer’s ability to implement one-step GD can be extended with FFN layer for projected GD or ISTA algorithm that can solve QP.
Based on this observation, authors test transformer architecture’s ability to perform portfolio optimisation, which is a common problem instance that can be solved utilising quadratic programming.
By augmenting the sample covariance to the intermediate token, authors found out that while predict-then-optimise tactic performs best in low transaction cost limit, but as limit relaxes, authors’ methodology perform better.

## Strength

Authors utilise the well-known result that transformers can perform GD in one layer, and extend such an idea to implement variants of GD that can be used to solve quadratic programming introduced in the paper.
Together with empirical verification of the author’s claim, their result is reasonable.

Authors also show practical application of their idea on portfolio optimisation, which requires the solver to utilise the covariance of the assets to correctly reflect the risk.
While the suggested methodology does not outperform throughout the task, it clearly shows benefits in weaker constraints.



## Areas for Improvement

While authors show that transformers can learn how to utilise covariance and solve quadratic programming, it doesn’t mean they can learn general, universal quadratic programming solvers.
As authors tokenise the QP problem into $n$ dimensional vectors of length $n+m+3$, the number of variables is upper bounded by the embedding dimension strictly.
Similarly, while the number of constraints allows generalisation as it only affects the sequence length, as Table 8~15 in supplementary material indicates (optimal configuration changes for different n and m pairs), the transformer can only solve a single (n, m) pair.
This would remain in the practical setting, if one tries to expand the assets of interest, one requires to retrain the transformer.

Also, authors construct the transformer weights to show learnability of transformers for specific algorithms, but this does not imply transformers can and will learn such algorithms.
To justify such claims, theoretically, one can consider small models as one-layer model (https://arxiv.org/abs/2306.09927) or fewer parameter models (https://arxiv.org/abs/2306.00297) and analyse training dynamics or loss landscape, show specific algorithms can be reached with GD.
Empirically, one can consider more systematic study, comparing how similar algorithmic solution and transformer behaves differing hyperparameters as depth or iteration step (https://arxiv.org/abs/2310.17086).

## Detailed Comments

The two regimes that Predict-then-Optimize and Time2Decide can be interesting.
Do you have any idea on why this is happening? One possibility is the projection algorithm to satisfy constraint. Time2Decide might fail to satisfy strong constraints (small $\gamma$) and while projection handles constraint invalidation, it can lose precision during the prediction, where Predict-then-Optimize doesn’t.
It would be helpful if authors report the invalidation ratio or average projection length.

Also, reporting the MSE between optimal allocation and predicted allocation seems misaligned to the problem.
As risk can be computed given allocation, it would be better to report the loss ($ s_t^\intercal r_t - \lambda s_t^\intercal \hat{\Sigma}_t s_t$), or each terms in the loss, risk and return.

## Justification of Score

While authors introduce possible directions that transformers can handle matrix valued instances, current methodology does not seem to utilise the structure of matrix and transformer well, and theoretical results presented are insufficient to argue that transformers will learn QP solving algorithms.
The empirical results on portfolio optimisation is, while correctly designed, and shows some interesting behaviour when ICL is applied to constrained problem, but is not the scope of paper for now.

---

### Official Review · Reviewer_PVkU · 2026-05-22

**Rating:** 6
**Confidence:** 4

**Review:**

## Summary

The paper investigates the capability of transformer architectures to solve Quadratic Programs (QPs) and applies this to downstream decision-making tasks. The authors show that Linear Transformers outperform Softmax Transformers in learning to approximate QP solutions. Then, the paper introduces Time2Decide, a method that augments Time-Series Foundation Models (TSFMs) with explicit covariance tokens to solve portfolio optimization problems end-to-end, outperforming a classical Predict-then-Optimize (PtO) baseline on synthetic data. In addition, they theoretically demonstrate how specific transformer components, such as linear attention blocks and Feed-Forward Networks, can explicitly emulate classical first-order optimization algorithms (e.g., Gradient Descent for unconstrained QPs, or the Arrow-Hurwicz method for linearly constrained QPs).

## Strengths

- The idea to introduce Covariance-Augmented Tokenization (CAT) is very interesting and highly practical. This enables the repurposing of a time-series forecasting foundation model into an end-to-end model for portfolio optimization.
- Overall, the paper is technically sound without major flaws in its core methodology.
- The theoretical derivations linking transformer components to iterative optimization algorithms are highly promising, though I did not review them exhaustively as they were relegated entirely to the appendix.

## Areas for Improvement
- The paper reads like a condensed conference submission, resulting in pacing and structural issues. The introduction and conclusion are overly long and attempt to sell all contributions, while the theoretical results, which could potentially be the primary contribution, are hidden entirely in the appendix. (Note: the appendix section "Proofs of Main Results from Section 4" references a Conclusion section, highlighting this condensation issue). I recommend that the authors significantly rework the introduction and conclusion, move a high-level summary of the theoretical proofs into the main text, and better explain the Time2Decide architecture.
- The related work should be significantly improved in two key areas: (1) The forecasting section cites mostly older models (Autoformer, FEDformer, Crossformer), while overlooking recent time-series foundation models (e.g., Chronos, Toto, MOIRAI). (2) The authors do not contextualize their linear attention contribution within the broader literature on attention expressivity. They should consider citing foundational linear attention work (e.g., Katharopoulos et al., 2020) and recent theoretical papers analyzing the expressive differences between softmax and linear attention (e.g., Deng et al., 2024; Han et al., 2024).
- Evaluating a portfolio optimization model purely on Mean Squared Error (MSE) to an oracle allocation is mathematically convenient but practically insufficient for financial applications. The authors should evaluate the downstream financial utility of their allocations by reporting standard metrics like the Sharpe ratio, Cumulative Return, Maximum Drawdown, and actual Turnover Costs.
- The portfolio optimization experiments are conducted exclusively on synthetic multivariate time series. While this allows for controlled experiments, evaluating Time2Decide on real-world financial datasets (e.g., historical stock market returns) would significantly strengthen the claim that end-to-end models practically outperform Predict-then-Optimize methods in noisy, real-world environments.

*References.*

(Katharopoulos et al., 2020) Transformers are RNNs: Fast Autoregressive Transformers with Linear Attention

(Han et al., 2024) Bridging the Divide: Reconsidering Softmax and Linear Attention

(Deng et al., 2024) Why Softmax Attention Outperforms Linear Attention

## Detailed Comments

- You note that the Linear Transformer outperforms the Softmax Transformer, and Appendix J provides a great theoretical obstruction regarding the negativity/even-degree limitation of pure data tokens in linear attention. However, as noted above, it would be better to contextualize this within the existing literature, which usually states that softmax attention has superior expressive power (Deng et al., 2024; Han et al., 2024).
- During the Time2Decide training, the model outputs raw, unconstrained vectors and relies on a solver for post-hoc projection during inference. Have you considered incorporating a differentiable simplex projection directly into the architecture's output head? Using a Softmax layer (with a learnable temperature) or a Sparsemax projection would guarantee that the non-negativity and budget constraints are satisfied natively during the forward pass. This would likely stabilize training and reduce reliance on external solvers.
- Section 3.2 mentions applying a "unified positional encoding to the entire sequence." If the sequence contains a concatenated mix of structural tokens (covariance) and temporal tokens (historical returns), a unified 1D positional encoding destroys the temporal inductive bias. Please clarify your PE strategy; utilizing a 2D positional encoding or separate embedding spaces for cross-sectional vs. sequential data would be more appropriate.
- In Section 3.2, "Pretrained" is described as "TimePFN used purely as a neural forecaster" (implying it outputs returns). However, in Appendix E.2.2, it is grouped under "End-to-End Neural Strategies" that "generate allocations directly." If it is a zero-shot forecaster, it cannot directly output portfolio allocations without a trained projection head. If it does have a trained head for allocations, how does it differ from the SFT baseline? Please clarify.
- If I understand correctly, the CAT idea fits any patch-based transformer model. Trying another foundation model as the base for Time2Decide would significantly strengthen the empirical claims regarding its generalizability.
- In Section 2.1, $m$ is used to denote the number of linear inequality constraints ($n=5, m=3$). In Section 3.1, $m$ is used to denote the number of financial assets ($m=16$). Please use distinct variables to prevent reader confusion.


## Justification of Score
The paper makes a compelling, technically sound argument bridging continuous optimization theory with transformer capabilities, and the proposed Time2Decide architecture demonstrates clear utility. However, the paper currently suffers from structural issues stemming from its conversion to a workshop format, misses important related work, and lacks a more realistic experimental setup. Overall, I recommend acceptance, conditioned on the resolution of the formatting issues, refinement of the related work, and explicit discussion of the empirical limitations.